# INC: An Indirect Neural Corrector for Auto-Regressive Hybrid PDE Solvers

**Hao Wei**
Technical University of Munich
`hao.wei@tum.de`

**Aleksandra Franz**
Technical University of Munich
`aleksandra.franz@tum.de`

**Bjoern List**
Technical University of Munich
`bjoern.list@tum.de`

**Nils Thuerey**
Technical University of Munich
`nils.thuerey@tum.de`

## Abstract

When simulating partial differential equations, hybrid solvers combine coarse numerical solvers with learned correctors. They promise accelerated simulations while adhering to physical constraints. However, as shown in our theoretical framework, directly applying learned corrections to solver outputs leads to significant autoregressive errors, which originate from amplified perturbations that accumulate during long-term rollouts, especially in chaotic regimes. To overcome this, we propose the Indirect Neural Corrector (INC), which integrates learned corrections into the governing equations rather than applying direct state updates. Our key insight is that INC reduces the error amplification on the order of $\Delta t^{-1} + L$, where $\Delta t$ is the timestep and $L$ the Lipschitz constant. At the same time, our framework poses no architectural requirements and integrates seamlessly with arbitrary neural networks and solvers. We test INC in extensive benchmarks, covering numerous differentiable solvers, neural backbones, and test cases ranging from a 1D chaotic system to 3D turbulence. INC improves the long-term trajectory performance ($R^2$) by up to 158.7%, stabilizes blowups under aggressive coarsening, and for complex 3D turbulence cases yields speed-ups of several orders of magnitude. INC thus enables stable, efficient PDE emulation with formal error reduction, paving the way for faster scientific and engineering simulations with reliable physics guarantees. Our source code is available at `https://github.com/tum-pbs/INC`

## 1 Introduction

Numerical simulation of nonlinear partial differential equations (PDEs) over long time horizons is crucial for many scientific and engineering scenarios, with applications ranging from sub-grid turbulence closure in climate models to real-time aerodynamic design and plasma physics. Traditional *numerical solvers* deliver rigorous solutions [1–3], but their computational cost becomes prohibitive when resolving fine-scale features. Designed to replicate the behavior of traditional numerical solvers, *neural emulators* seek to accelerate solving by learning a surrogate directly from data [4–14]. However, their long-term performance is problematic, especially for chaotic dynamics, as a tiny inaccuracy from a single step of an emulator can lead to solutions drifting out of distribution [15–18].

On the other hand, *hybrid neural solvers* combine benefits from numerical and neural methods by integrating a coarse-grid physics solver as a backbone with a neural network. This network reconstructs missing, unresolved, or erroneously represented physics, akin to closure models [19]. Such approaches can be broadly categorized into two types: *learned correction* and *learned interpolation*.

39th Conference on Neural Information Processing Systems (NeurIPS 2025).

In learned interpolation [20, 21], a discretization scheme is directly learned on a coarse grid. This approach modifies the solver at the level of numerical discretization to mimic the effect of closures [22]. In contrast, learned correction targets unresolved physics through *direct* correction terms, offering a simpler implementation and flexibility [23–30]. The **directness** of the existing learned correction methods arises from applying a learned correction term directly to the output of a coarse-grid solver after each time step. Generally expressed as $u^{n+1} = u^{n^*} + \mathrm{LC}(u^{n^*})$, where $u^{n^*}$ is the solution from the embedded solver, learned correction methods show improved accuracy in rollout trajectories. Still, hybrid solvers are prone to instabilities caused by perturbations. Such instability has long been recognized in multi-physics solvers [31], and stability can further deteriorate when coupling numerical solvers with neural networks [22, 23]. Worse still, existing stabilization strategies for pure neural emulators fail here. Strategies with changing timescales can violate the strict stability criteria of solvers, and noise injection only leads to subtle improvements [23]. While multi-step rollout strategies have been applied and were shown to be helpful [18, 23, 27], the marginal stabilization doesn't change the intrinsic sensitivity to perturbations of direct correction. This inherently limits the application of existing hybrid frameworks, highlighting important shortcomings of existing methods.

To address these shortcomings, we first develop a theoretical *error* growth framework for hybrid neural solvers under autoregressive rollout. We find that for direct learned correction, local perturbations are amplified with $G(u^n) = I + \Delta t J(u^n)$, where $J(u^n)$ is the Jacobian of the system evaluated at $u^n$. Contrarily, if the learned correction is applied in an **indirect** way, e.g., via a right-hand-side (RHS) term, local perturbations scale only with $\Delta t$. In autoregressive rollouts, the difference of these two corrections grows with a ratio $R_k \sim \Delta t^{-1} + L \gg 1$, where $L$ is the Lipschitz constant. Motivated by this result, we introduce the *Indirect Neural Corrector* (INC), a hybrid solver paradigm that (1) embeds learned corrections within the per-time-step PDE solve, (2) provably reduces *errors* with scaling $\frac{1}{R_k}$, where $\frac{1}{R_k} \ll 1$, and (3) is compatible with arbitrary neural architectures and numerical schemes.

To demonstrate the practical impact of INC, we conduct extensive benchmarks on six canonical PDE systems and varying differentiable solvers as well as neural network architectures. INC consistently outperforms direct hybrid and pure-neural baselines, strongly reducing long-term trajectory errors, and yielding speed-ups of up to $330\times$ for complex 3D turbulence cases. Our main contributions are summarized as follows.

- **A theoretical framework.** We derive a unified error-propagation framework for hybrid neural solvers under autoregressive rollout, and show that indirect correction reduces the worst-case drift by $\mathcal{O}(R_k)$ compared to direct methods.

- **INC algorithm and architecture.** We design a general indirect correction module that integrates seamlessly with off-the-shelf numerical solvers and neural networks, imposing stability constraints without sacrificing expressivity.

- **Extensive benchmarks.** We evaluate with 6 canonical PDE systems (from 1D chaotic system to 3D turbulence), 4 architectures (FNO, DeepOnet, ResNet, U-Net), and 3 differentiable solver types (finite-volume, finite-difference, pseudo-spectral).

## 2  Related Work

**Neural Emulators**  Neural emulators seek to accelerate solving PDEs by learning a surrogate directly from data [7–10], with minimal additional constraints like a physical loss [11, 12] and symmetries [8, 13], or design the architecture to mimic the numerical solver [4, 13, 14]. While these models can substantially reduce per-step cost, they generally lack stability guarantees, leading to trajectory divergence [32] and unphysical statistics during inference [33]. Especially for chaotic dynamics with exponentially growing perturbations, a tiny inaccuracy from the single network application can lead to solutions drifting out of distribution [15–18]. Existing remedies, like multi-step rollouts during training [34, 35], multi-model training with multi-timescales [36], noise injection [5, 32], learning timescales [6], incorporating knowledge like invariant measure [17, 37] and multi-resolution targets [12] reduce errors at the cost of increased complexity and hyperparameter tuning. More importantly, these methods still provide no formal stability guarantee.

**Hybrid Neural Solvers**  Hybrid neural solvers couple a solver with a neural networks, where the network learns an interpolation [20, 21] or a correction [23–30] This approach has been applied in

various domains including vortex shedding [23], flow control [28], and turbulence modelling [21, 25, 26]. Recent advancements include extensions with online learning [38] and physical losses [27]. Previous works share some similarities with our approach [26, 27, 39]. However, CoSTA [39] is limited to the training of a DNN for single-step corrections on a 1D diffusion problem. List et. al [27] only focused on correction in advection steps, where a learned correction might be regularized by following pressure-solves. Lastly, in the case of DPM [26], the use of a simple MLP and the absence of a modern ML framework for the solver limit the augmentation between the solver and NN, making it difficult to integrate with state-of-the-art models. While all the above-mentioned studies focus on building a closure model, they do so without the theoretical analysis of the growth of perturbations in hybrid solvers that we provide in this paper.

## 3 Method: Indirect Neural Correction

Solver-in-the-loop style approaches, which we term *direct* corrections in this work, take an operator-splitting approach and correct a coarse numerical solution with a neural network

$$u^* = \mathcal{T}\big[u^*, u^n, \mathcal{N}(\partial_x u^n, \partial_{xx} u^n, \dots)\big], \quad u^{n+1} = \mathcal{G}_\theta(u^*), \tag{1}$$

where $\mathcal{T}$ represents the (explicit or implicit) temporal integration, $\mathcal{N}(\cdot)$ denotes the physical dynamics governed by spatial derivatives, and $\mathcal{G}_\theta(u)$ is a neural network parameterized by $\theta$. Thus, direct approaches compute a correction to the dynamics directly as a state update.

In contrast, an *Indirect Neural Correction* (INC) performs corrections as a RHS term in the underlying PDE

$$u^{n+1} = \mathcal{T}\big[u^{n+1}, u^n, \mathcal{N}(\partial_x u^n, \partial_{xx} u^n, \dots) + \mathcal{G}_\theta(u^n)\big]. \tag{2}$$

When neural corrections introduce small perturbations, this subtle difference of moving $\mathcal{G}_\theta$ into the time integration can have stark effects on the stability of inference rollouts, as our theoretical analysis in section 4 and the results in section 6 show. Note that the differentiability of $\mathcal{T}$ is required to optimize $\theta$ for the INC case, even when only unrolling one step.

We train the models in a supervised fashion based on high-resolution data. To ensure that the model learns to produce stable and accurate long-term predictions, we employ a multi-step unrolled optimization strategy. The training objective is typically defined via an $\mathcal{L}^2$ loss:

$$\theta^* = \arg\min_\theta \left[ \sum_{t=n}^{N-m} \sum_{s=1}^{m} \mathcal{L}_2\left(\tilde{u}^{n+s}, (\mathcal{T}|\mathcal{N}|\mathcal{G}_\theta)^s(\tilde{u}^n)\right) + \lambda\|\theta\| \right], \tag{3}$$

where $\tilde{u}^n$ denotes the filtered high-fidelity solution, $\lambda$ enforces Lipschitz continuity, and $(\mathcal{T}|\mathcal{N}|\mathcal{G}_\theta)^s$ denotes the autoregressive application of $s$ hybrid time steps, either following eq. (1) for direct corrections or eq. (2) for indirect ones. At inference time, the same $(\mathcal{T}|\mathcal{N}|\mathcal{G}_\theta)$ step is applied.

The temporal and spatial discretizations are implemented as a differentiable solver $\mathcal{S}$. The predicted trajectory is then generated by setting $\tilde{u}^n$ as the initial condition and recursively applying $\mathcal{S}$ with correction term $\mathcal{G}_\theta$ for $m$ steps:

$$\mathcal{S}^m\left(\tilde{u}^n, \mathcal{G}_\theta^s(\tilde{u}^n)\right) = \underbrace{\mathcal{S} \circ \cdots \circ \mathcal{S}}_{m \text{ times}}\left(\tilde{u}^n, \mathcal{G}_\theta(\tilde{u}^n)\right). \tag{4}$$

The neural correction $\mathcal{G}_\theta$ thus indirectly influences the solution through the solver, aligning the predictions with the ground-truth evolution governed by the PDE.

## 4 Theoretical analysis: Propagation of perturbations through the solver

In this section, we show how indirect corrections reduce the *error* by $\mathcal{O}(R_k)$ compared to direct corrections. We start by deriving a unified error-propagation framework for hybrid neural solvers under autoregressive rollouts. Subsequently, an error-dominance ratio ($R_k$) has been defined and connected to the timescale and Lipschitz bound, thereby quantifying the benefits of indirect approaches.

**Problem 4.1** (Perturbation Analysis). We consider a general PDE format:

$$\partial_t u = \mathcal{N}(\partial_x u, \partial_{xx} u, ...), \qquad (t, \mathbf{x}) \in [0, T] \times \mathbb{X} \tag{5}$$

$$u(0, \mathbf{x}) = u^0(\mathbf{x}), \quad \Omega[u](t, \mathbf{x}) = \mathcal{B}(u), \qquad \mathbf{x} \in \mathbb{X}, \quad (t, \mathbf{x}) \in [0, T] \times \partial\mathbb{X} \tag{6}$$

where $u(t, \mathbf{x})$ is the solution in $d$ spatial dimensions $\mathbf{x} = [x_1, x_2, ..., x_d]^T \in \mathbb{X} \subset \mathbb{R}^{n \times d}$ over time $t \in [0, T]$. $\mathcal{N}$ represents the spatial differential operator. $u(0, \mathbf{x})$ is initial condition and boundary condition described by function $\mathcal{B}$.

Now let us define the perturbations caused by neural network correctors in order to study their growth during autoregressive inference. The correctors either perturb the state (direct correctors) or the RHS of the equation (indirect correctors). Under the well-posedness, numerical stability, and small perturbation assumptions (detailed in appendix A), we define:

**Definition 4.1** (Perturbations and Errors). Let $\epsilon_u$ (direct perturbation) and $\epsilon_s$ (indirect perturbation) perturb the system as

$$u^{n+1} + \delta u^{n+1} = \mathcal{T}[u^{n+1}, \mathcal{N}(u^n + \epsilon_u^n) + \epsilon_s^n], \tag{7}$$

where $\mathcal{T}$ is any time-discretization, $u^{n+1}$ is the exact solution at step $n + 1$, and $\delta u^{n+1}$ is the resulting error due to the perturbations at step $n$. Both perturbations satisfy $\|\epsilon_u\| = \|\epsilon_s\| \leq \epsilon$, and are sufficiently small such that $\mathcal{O}(\epsilon^2)$ terms are negligible in Taylor expansions.

**Proposition 4.1** (Local Perturbations Propagation). *The error $\delta u^{n+1}$ at step $n + 1$, arising from perturbations $\epsilon_u^n$ and $\epsilon_s^n$ at step $n$, is given by:*

$$\delta u^{n+1} = (I + \Delta t J(u^n))\epsilon_u + \Delta t \epsilon_s = G(u^n)\epsilon_u + \Delta t \epsilon_s, \tag{8}$$

*where $J(u^n) = \frac{\partial \mathcal{N}}{\partial u}\big|_{u^n}$ is the Jacobian of $\mathcal{N}$ evaluated at $u^n$ and the $G(u^n) := I + \Delta t J(u^n)$, is the error amplification matrix. Proof see appendix B.1.*

Based on the propagation of local perturbations, we can derive the expression for the cumulative error after $k$ steps as:

**Lemma 4.1** (Cumulative Error). *After iterating eq. (8) for $k$ steps which introduce new perturbations $\epsilon^m$ at every step $m \in [0, k]$, the total error satisfies:*

$$\delta u^{n+k} = \sum_{m=0}^{k-1} \left( \prod_{i=m+1}^{k-1} G(u^{n+i}) \right) \left[ G(u^{n+m})\epsilon_u^m + \Delta t \epsilon_s^m \right], \tag{9}$$

Detailed derivation is provided in appendix B.2. To compare different effects of errors perturbed by $\epsilon_u$ and $\epsilon_s$, we define the error dominance ratio $R_k$ by setting $\epsilon_s = 0$ or $\epsilon_u = 0$ in eq. (9):

**Definition 4.2** (Error Dominance Ratio).

$$R_k = \frac{\|\delta u^{n+k}\|_{\epsilon_s=0}}{\|\delta u^{n+k}\|_{\epsilon_u=0}} = \frac{\left\| \sum_{m=0}^{k-1} \left( \prod_{i=m+1}^{k-1} G(u^{n+i}) \right) G(u^{n+m})\epsilon_u^m \right\|}{\left\| \sum_{m=0}^{k-1} \left( \prod_{i=m+1}^{k-1} G(u^{n+i}) \right) \Delta t \epsilon_s^m \right\|} \tag{10}$$

**Remark 4.1.** The products in eq. (10) describe the propagation of perturbations through long unrolled trajectories. The outer sums in turn represent the introduction of new perturbations at every timestep. Depending on the characteristics of the underlying dynamics, either term can dominate.

**Proposition 4.2.** *Under the assumption that the nominator and denominator of eq. (10) grow like their upper bounds, the ratio $R_k$ can be simplified to:*

$$R_k \sim \Delta t^{-1} + L \gg 1. \tag{11}$$

*where $L$ is the Lipschitz bound. Proof see appendix B.3.*

**Remark 4.2.** The ratio $R_k$ relates the growth of direct and indirect perturbations. Since $L \geq 0$ by definition and $\Delta t \ll 1$ in general, one can state that $R_k \sim \Delta t^{-1} \gg 1$ in most practical applications. The growth of the error in an unrolled trajectory is thus dominated by the perturbation $\epsilon_u$.

An even stronger statement can be made for chaotic dynamics, where the growth of perturbations can be related to Lyapunov stability.

**Theorem 4.1** (Lyapunov Stability). *The Lyapunov stability framework characterizes the response of dynamical systems to perturbations. A chaotic PDE has a positive maximum Lyapunov exponent $\lambda_{max} > 0$. For a trajectory $u(t)$, it is defined as:*

$$\lambda_{max} = \limsup_{t \to \infty} \frac{1}{t} \ln \frac{\|\delta u(t)\|}{\|\delta u(0)\|}. \tag{12}$$

**Proposition 4.3** (Lyapunov Stability with upper-bounds). *In current linearized dynamics analysis, the Lyapunov exponents are determined by $J(u)$ and bounded by $L$. Specifically, $\lambda_{max} \leq L$. Proof can be found in appendix B.4. Thus, $L$ upper-bounds $\lambda_{max}$ with $L > max(0, \lambda_{max})$.*

**Remark 4.3** (Numerical Implications). In stable systems with ($\lambda_{\max} < 0$), the error ratio $R_k$ is not affected by the Lyapunov exponent and is bounded by the non-negativity of $L$. The effects detailed in the previous remark 4.2 remain. In chaotic system with $L \geq \lambda_{\max} > 0$, the error ratio follows $R_k \sim \Delta t^{-1} + L$, implying that direct perturbation errors $\delta_u$ grow far more rapidly than indirect perturbation errors $\delta_s$. The exponential growth of perturbation in chaotic dynamics thus further amplifies the dominance of direct errors $\delta_u$.

Moreover, the amplified $\epsilon_u$-error may drive the numerical solution outside the neighborhood where the Lipschitz bound $\lambda_{\max} \leq L$ holds, risking a loss of convergence or blowup. This is further exacerbated for hybrid solvers where NNs may exhibit high sensitivity to input perturbations [40], so the compounded amplification from $R_k$ undermines both accuracy and stability of the learned model.

The above theoretical analysis is based on linearized error dynamics under the assumption of small perturbations. This assumption enables a tractable derivation and a clear understanding of how direct and indirect correction mechanisms differ in error-amplification. To address this theoretical limitation, on the one hand, we deploy a pure numerical study by introducing Gaussian noise as a perturbation at each time step via direct/indirect mechanisms as detailed in appendix C. On the other hand, we have extended our experimental setup to include various challenging scenarios, like chaotic and turbulent cases, which are detailed in section 6.

## 5   Experimental Configurations

We compare INC to three other architectures: SITL, SITL*, and RNN. Both SITL and SITL* are direct correction methods and only differ in the order in which the direct correction and the solver operate. For SITL, the correction is applied after the solver [21, 23]. Here, a coarse numerical solver $\mathcal{T}$ first advances the state, producing an intermediate solution $u_n^* = \mathcal{T}(u_n)$. Then, a neural network $G_\theta$ directly corrects the solver output to yield the next state: $u_{n+1} = G_\theta(u_n^*)$. This incorporates a residual connection such that $G_\theta(u_n^*) = u_n^* + \theta(u_n^*)$. SITL* is a simple pre-correction variant of SITL. Instead of adding the learned correction after the solver update, SITL* applies the neural term before the solver step. First, the network calculates $u_n^* = G_\theta(u_n) = u_n + \theta(u_n)$, which is then used as input to the coarse solver such that $u_{n+1} = \mathcal{T}(u_n^*)$. We also compared our work with CSM [24] in appendix I. CSM is a variant of SITL that scales the correction with the time step. By evaluating SITL, SITL*, and CSM, we ensure that our work encompasses the entire landscape of direct methods. Furthermore, the implementation of the RNN follows the FNO architecture [4]. It takes a sequence of past solution states $(u_{n-T+1}, \ldots, u_n)$ and predicts the next state $u_{n+1}$ while operating autoregressively. Herein, following FNO, we set $T = 10$. In contrast to the other approaches, the RNN is a solver-free and fully data-driven transition operator; other setups are kept identical to a hybrid solver, relying on learned temporal dependencies through autoregression.

We employ *unrolling* multiple time steps at training time [18, 21, 23]. It improves inference performance and increases the memory requirements since it is necessary to store neural network activations inside each unrolled time step in the forward pass. In our work, the choice of unrolling length is guided by the characteristic timescale $t_c$ of the dynamical systems. For each case, $t_c$ is derived from system-specific properties, like the inverse of the maximum Lyapunov exponent for the KS equation. More details about $t_c$ for other cases are listed in appendix H. After calculation of $t_c$ and setting the time step as $\Delta t$, we compute the number of steps per characteristic timescale as $N = \frac{t_c}{\Delta t}$. We then set the unrolling length during training to approximately $0.04 \cdot N$. During inference, we significantly extend this horizon to approximately 100 times longer, typically $4 \cdot N$ to $6 \cdot N$, to rigorously evaluate long-term stability and accuracy.

We comprehensively evaluate INC across 3 distinct numerical solvers, each representing a canonical class of spatial discretizations and temporal schemes:

- A finite-difference method (FDM) solver for the Burgers equation, combining fifth-order WENO spatial reconstruction with explicit forward Euler time integration [1].
- A Fourier pseudo-spectral solver (PS) for the KS equation, employing the exponential time differencing Runge-Kutta scheme (ETDRK) [2].

- A finite-volume method (FVM) solver for the incompressible Navier-Stokes equations, leveraging the semi-implicit PISO algorithm for pressure-velocity coupling [3].

While INC universally corrects solutions by integrating $\mathcal{G}_\theta$ in the RHS of the governing equations, its mechanism for integration varies slightly across temporal schemes. For forward Euler steps, $\mathcal{G}_\theta$ is appended to the source term and advanced in time via first-order explicit integration. For ETDRK, $\mathcal{G}_\theta$ is fused with the nonlinear term in Fourier space during the ETD integration step, inheriting the scheme's exact treatment of stiff linear operators while retaining spectral accuracy. For PISO, $\mathcal{G}_\theta$ is embedded within the momentum equation's source term, influencing both the velocity predictor step and the iterative pressure correction process continuously.

We deploy two inherently different types of neural networks: convolution-based architectures like ResNet and UNet, as well as FNO and DeepONet as operator-based ones. We demonstrate the accuracy and long-term stability of our methods on multiple tasks with varying difficulty and with different numerical schemes. Details for each case and solver type are provided in the appendix H.

**Kuramoto–Sivashinsky (KS) Equation**   The KS equation is a well-known chaotic system modelling intrinsic instabilities, such as reaction-diffusion systems [41, 42]. It can be written as: $\frac{\partial u}{\partial t} + u\frac{\partial u}{\partial x} + \frac{\partial^2 u}{\partial x^2} + \frac{\partial^4 u}{\partial x^4} = 0$. The interplay between the contrasting terms leads to significant spatio-temporal complexity [43], sensitive to domain length $L$. When $L$ varies, the KS equation produces distinctly different dynamics [18, 41, 42]. A Pseudo-Spectral method with Exponential Time Differencing [2] was applied, where the non-linear terms are treated with a second order Runge-Kutta time integration [44, 45] with $\Delta t = 0.01$. The spatial resolution was fixed to 64.

**Burgers' Equation**   The Burgers' equation is a nonlinear advection–diffusion PDE with shock formation: $\frac{\partial u}{\partial t} + u\frac{\partial u}{\partial x} = \nu\frac{\partial^2 u}{\partial x^2} + \delta(x, t)$. We use a scenario with forcing $\delta(t, x) = \sum_{j=1}^{J} A_j \sin(\omega_j t + 2\pi\ell_j x/L + \phi_j)$. The setup follows Bar-Sinai et al. [20], where $J = 5$, $L = 16$, $A_j \sim \mathcal{U}[-0.5, 0.5]$, $\omega_j \sim \mathcal{U}[-0.4, -0.4]$, $\ell_j \in \{1, 2, 3\}$, and $\phi_j \in [0, 2\pi]$. The domain is discretized on $n_x = 512$ with periodic boundary and advanced explicitly in time with $\Delta t = 10^{-3}$. A solver based on WENO-5 has been applied to capture shocks.

**Navier–Stokes (NS) Equation**   The Navier–Stokes equations describe the motion of fluids and are fundamental in modeling phenomena such as turbulence, boundary layer flows, and vortex dynamics [46, 47]. Herein, we consider the incompressible NS equation, which can be expressed as $\frac{\partial u}{\partial t} + (u \cdot \nabla)u = -\nabla p + \nu\nabla^2 u + \mathbf{f}$ with $\nabla \cdot u = 0$ for conservation of mass. Here $u$ denotes the velocity field, $p$ is the pressure, $\nu$ represents the kinematic viscosity, and $\mathbf{f}$ is external forcing. The divergence-free condition ensures the incompressibility of the flow. We used a differentiable solver based on the 2nd-order PISO algorithm with a spatially adaptive finite-volume discretization. To comprehensively study INC, we evaluate three different turbulent cases, 2D Karman vortex shedding (Karman), 2D Backward facing step (BFS) and a turbulent channel flow (TCF) in 3D. All cases, and especially the 3D TCF case, feature complex dynamics and a large number of degrees of freedom.

## 6   Results

In the following, we evaluate INC with tests designed to investigate long-term accuracy, solver stability under unstable conditions, and applicability to accelerate complex turbulent cases.

### 6.1   Accuracy in Long Autoregressive Rollouts

To evaluate INC's ability to suppress perturbation growth during autoregressive rollout while maintaining high fidelity, we test it with the chaotic KS equation (denoted by KS1) and the shock-forming Burgers case. Both inference simulations are conducted with a fine timescale and a large number of steps, 4000 and 5000 for Burgers and KS. [1] To compare, we benchmark against a classic direct approach SITL [23], a pre-correction variant SITL*, and a recurrent baseline without solver RNN following, e.g., Li et al. [4]. All variants, including INC, use identical NN architectures.

---

[1] The simulation times for KS and Burgers are approximately $4.5t_{c,KS}$ and $4t_{c,BG}$. The characteristic timescale $t_c$ reflects fundamental properties of the dynamical systems, as detailed in the appendix H.

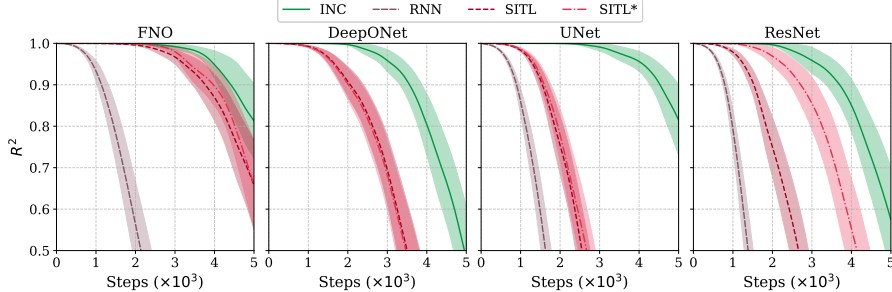

Figure 1: $R^2$ correlation of KS1 over 5000 steps for different methods (INC, RNN, SITL, SITL*) applied to four models (FNO, DeepONet, UNet, ResNet). The INC method consistently demonstrates the most stable long-term behavior, maintaining high accuracy even beyond 5000 steps.

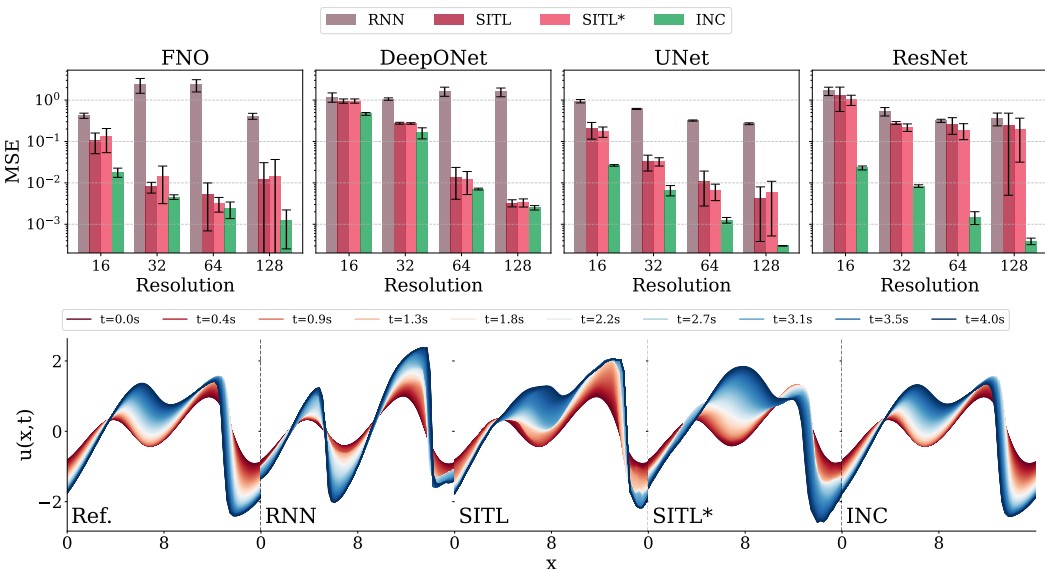

Figure 2: **Top** MSE of 4000-step Burgers rollouts across different neural architectures and spatial resolutions. **Bottom** Spatiotemporal evolution of the velocity field by different methods with 4000 steps. INC always performs the best

**Enhanced Accuracy for Chaotic Systems**   Chaotic systems exponentially amplify perturbations, causing pointwise divergence over the Lyapunov timescale. Thus, maintaining frame-by-frame trajectory accuracy ($R^2$) indicates strong perturbation control with reduced errors from each step. INC constantly performes better than direct corrections SITL and SITL*, as shown in fig. 1. Across the four architectures, the improvement in $R^2$ is up to 158.71% compared to SITL for a UNet architecture. For the FNO, INC only shows an improvement by ca. 3% over the SITL versions, which can be explained by the FNO's architecture being very similar to the underlying pseudo-spectral solver [33]. For DeepONet, which employs MLPs instead of convolutions like the other models, INC improves by 35.26% / 37.15% over SITL / SITL*.

**Enhanced Accuracy for Shock Waves with Reduced Resolution**   Reduced resolutions lower the computational costs but smear critical details, particularly for challenging discontinuities like shocks. Autoregressive perturbations can exacerbate errors in these setups, demanding robust correction methods. This sets the stage for testing INC, which consistently achieves the lowest MSE across all resolutions and neural architectures. Compared to both SITL and SITL*, INC reduces MSE by 39% to 99%, with the largest decrease seen against SITL with a ResNet. Fig. 2 further shows INC's ability to accurately reconstruct the detailed features of the flow field across 4000 steps, closely resembling the reference. Both SITL* and SITL methods, while capturing general trends, clearly

suffer from notable distortion, particularly evident at later time steps. These observations collectively highlight INC's capability to effectively address resolution-induced inaccuracies and autoregressive error propagation, preserving essential flow characteristics at reduced computational resolutions.

It is worth noting that across both KS1 and Burgers cases, the RNN approach consistently underperforms in comparison to the hybrid neural solvers in all metrics. This unsatisfactory performance aligns with known limitations of RNN-based methods, such as the absence of an embedded solver [18, 21], and their inherent sensitivity to gradient divergence in chaotic regimes [15]. Consequently, we exclude the RNN from subsequent, more challenging experiments. Since INC invariably outperformed other direct correction methods across all tested neural backbones, we will now fix the network architecture to focus on analyzing other aspect of the method. We use a convolution-based architecture in all following experiments, apart from the following KS case, where we employ an FNO due to its similarity to the pseudo-spectral solver used for KS.

## 6.2 Improving Numerical Stability

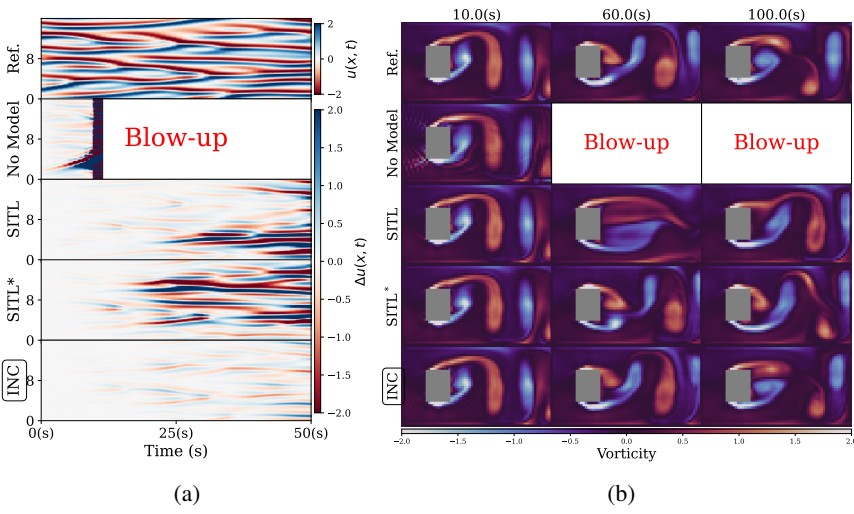

(a)          (b)

Figure 3: **(a)** KS2 simulation results with $L = 21.6\pi$, $dt = 0.5$s. Reference at the top, and errors of the no correction baseline and different hybrid solvers below. The solver (No Model) fails after 28 steps, while hybrid solvers (INC, SITL, SITL*) continue throughout the time horizon. **(b)** Simulated states for the vortex street case with $dt = 0.1$s. (No Model) diverges within 395 steps, while hybrid solvers (INC, SITL, SITL*) remain stable, with INC showing the best performance.

With INC effectively dampening perturbations through long-term rollout, we next assess its capability of improving numerical stability in two test cases where a coarse standard solver (No Model) fails.

**Low-order Temporal Schemes** When the numerical solver utilizes a simple low-order time integrator with a large time step, it can diverge rapidly. For 1D KS, with a first-order Runge-Kutta time integration with $\Delta t = 0.5$, the simulation blow-up typically occurs within $t \approx 14$s, as shown in fig. 3a. We denote this case by KS2. Under the same low-order time integrator, SITL and SITL* manage to reach $t = 50$s but with a degraded $R^2$ accuracy of 0.65 and 0.58. In

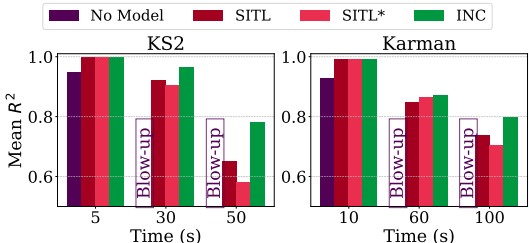

Figure 4: $R^2$ correlation over time ($\frac{1}{N}\sum_{i=1}^{N} R^2$) for different architectures (INC, RNN, SITL, SITL*) for both KS2 and Karman cases. All hybrid solvers improve stability compared to No Model, with INC giving the highest correlations.

contrast, INC attains a correlation of 0.78, representing a 20.0% improvement over SITL and a decrease by 36.9% in terms of MSE relative to SITL*, as detailed in fig. 4.

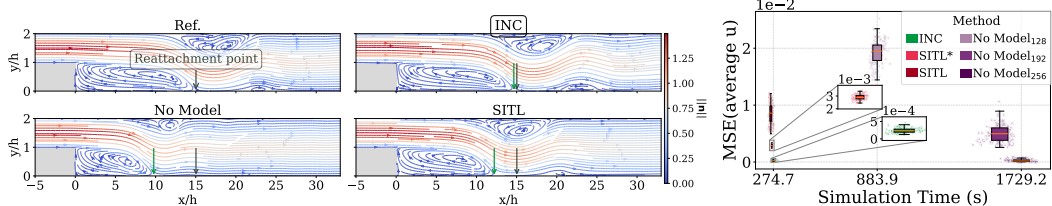

Figure 5: Left: BFS stream plots averaged over 1200 steps for Reference, INC, No Model, and SITL showing long-term emerging vortex structures in the flow. Right: efficiency-accuracy comparisons, INC shown in green. compared with numerical solvers of increasing resolution.

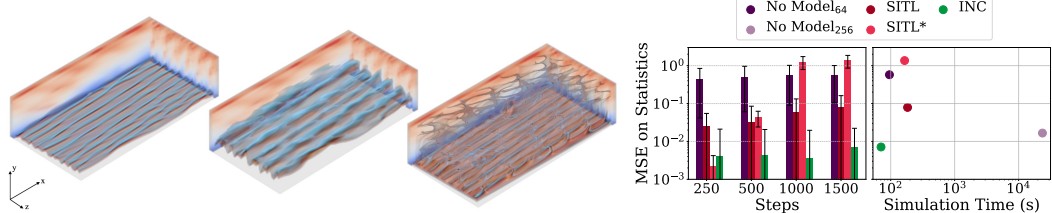

Figure 6: Right panels show three visualizations of states from the 3D TCF scenario with INC ($u_x$ near wall, $u_x$ bulk, $u_y$), while the two graphs on the left show accuracy over rollout steps and runtime. For the turbulent dynamics of this test case INC yields clear improvements on both fronts.

**Oversized time steps**  A No Model simulation diverges within 395 steps on average when the CFL number is set to 2.0, which is twice the conventional stability limit of CFL $< 1.0$. Combined with the spatial challenges caused by low resolution, this leads to a quick failure of the solver. SITL and SITL* manage to simulate 1000 steps but deliver only modest fidelity with an $R^2$ of 0.74 and 0.71. INC improves the $R^2$ to 0.80, and successfully reduces the MSE by 26.9% and 36.5%, respectively.

### 6.3   Acceleration and Accuracy Improvements for Complex Cases

Traditional solvers often impose strict discretization constraints to ensure stability and accuracy, which limits computational efficiency. Building on INC's accuracy in extended rollouts and stability in blow-up regimes, we further apply it to two turbulent flows with engineering relevance: a 2D BFS with 1200 rollout steps, and a 3D TCF with 1500 rollout steps.[1]

**Emerging Structures in Backward Facing Step**  We apply hybrid solvers to BFS with $4\times$ spatial and $20\times$ temporal down-sampling, resulting in CFL $= 4.0$. In contrast, the No Model requires adaptive time steps with CFL$< 0.8$ for stability. This scenario is challenging as accurately resolving the right vortex positions that emerge from the transient flow over longer intervals requires very high solving accuracies. The No Model and SITL baselines both fail to capture this behavior, as shown in comparison to a high-fidelity reference in fig. 5. The INC variant not only yields a stable simulation with a large CFL condition but also closely matches the emerging vortex structures.

The MSE of the averaged velocities confirms that INC achieves the highest accuracy, comparable to a high-fidelity No Model$_{256}$ simulation, which is conducted at approximately $2 \times$ the spatial resolution and $10 \times$ temporal resolution. Quantitatively, INC reduces the MSE by 97.1% compared to SITL. All hybrid solvers exhibit similar computational efficiency, achieving speedups of approximately 3 to $7\times$ compared to the No Model variants. As highlighted in the graph shown on the right of fig. 5, INC occupies the best position in the accuracy-performance Pareto front, yielding an accuracy that is on-par with No Model$_{256}$, while being $7\times$ faster.

**Matching Turbulence Statistics in 3D Flows**  For the 3D TCF case, featuring complex dynamics with a high number of degrees of freedom, we target a particularly challenging learning task: the hybrid solvers are trained to match turbulence statistics obtained from a high-fidelity spectral solver

---

[1]The simulation time is approximately $4t_{c,\text{BFS}}$ and $6t_{c,\text{TCF}}$ for BFS and TCF, as detailed in the appendix.

at high resolution [48], rather than reproducing individual grid values. Practical turbulent modeling prioritizes statistical consistency, e.g., mean velocity profiles and Reynolds stress, which characterize the average behavior of the flow and are robust to chaotic fluctuations [47]. For the TCF we employ these statistics as training loss instead of an $\mathcal{L}^2$ loss. Details are provided in appendix H.5.

The baseline methods, No Model, SITL, and SITL$^*$ employ a resolution of $64 \times 32 \times 32$ at a stable $CFL = 0.8$. INC utilizes the same spatial resolution but thanks to its stabilizing properties enables a $2\times$ time step, equivalent to $CFL = 1.6$. This CFL condition is unstable for the other baselines. Qualitative examples of an INC rollout, on the left side of fig. 6, show the intricate vortex structures forming in the volume.

Despite using larger timesteps, INC outperforms other methods, clearly occupying the best position in the accuracy-performance Pareto front shown on the right of fig. 6. While SITL$^*$ achieves a slightly better accuracy initially, it diverges significantly over time, reflecting its intrinsic sensitivity to perturbations during rollouts. Quantitatively, INC reduces the MSE on statistics by 80.7% compared to SITL. To achieve a comparable statistical accuracy without learned corrections, a significantly finer resolution of $256 \times 128 \times 128$ under extreme numerical stability constraints ($CFL= 0.1$) is necessary. This corresponds to a $4^3\times$ increase in spatial, and approximately $50\times$ temporal refinement. Consequently, even when running both solvers with the same GPU support and hardware, INC achieves a $330\times$ speed up compared to No Model$_{256}$ at equivalent statistical accuracy levels.

## 7   Concluding Remarks

In this work, we have introduced a unified theoretical framework to characterize error propagation in hybrid neural–physics solvers under autoregressive rollout, showing that direct correction terms amplify perturbations fundamentally faster than perturbations of indirect corrections. The resulting INC methodology leads to significantly reduced error growth, enhances accuracy, and maintains long-term stability across diverse PDE systems, architectures, and solver types. As a consequence, it achieves substantial improvements in predictive performance and computational efficiency for learned hybrid solvers.

While INC broadly enhances stability and accuracy, its advantage hinges on the dynamical stiffness of the PDE and the time-integrator's inherent stability region. In problems that are already well-damped or when employing unconditionally stable implicit schemes with large time steps, the relative benefits can diminish. Looking ahead, a highly interesting avenue for future work will be to investigate optimal strategies to choose rollout length, trading computational cost against learning effectiveness. Overall, our work makes an important step towards stable and accurate hybrid solvers, and applying it in real-world applications ranging from civil and maritime engineering to medical applications poses highly promising directions.

## 8   Acknowledgments

The authors are grateful for constructive discussions with Rene Winchenbach, Luca Guastoni, Mario Lino, Felix Köhler, Chengyun Wang, Yunjia Yang, Qiang Liu, Patrick Schnell, and Xiyu Huang.

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

# Appendix

Below, we present the complete theoretical developments referenced in the main text, beginning with the formal assumptions in appendix A and continuing with the detailed proofs provided in appendix B. These are followed by a numerical perturbation study in appendix C, which complements our theoretical analysis. In appendix D, we provide an in-depth description of our proposed method, Indirect Neural Correction (INC), including a schematic illustration that clarifies its core mechanisms. Subsequently, we outline the differentiable solvers used to implement the INC framework. Specifically, we detail the use of the Weighted Essentially Non-Oscillatory (WENO) scheme for the Burgers equation in appendix E, the Fourier pseudo-spectral method in appendix F, and the Pressure Implicit with Splitting of Operators (PISO) algorithm in appendix G. Finally, we provide additional implementation details and hyperparameters for all experiments in appendix H.

## A  Assumptions

In this section, we provide the assumptions for the theoretical framework.

**Assumption A.1** (Well-posedness). *The initial value problem for the PDE is well-posed in $H^s(\mathbb{X})$, the Sobolev space of order $s$. Specifically:*

- *Existence and Uniqueness: For every initial condition $u(0) \in H^s(\mathbb{X})$, there exists $T > 0$ and a unique solution $u \in C([0, T]; H^s(\mathbb{X}))$.*

- *Continuous Dependence: The solution map $u^0 \mapsto u(t)$ is continuous in the $H^s$-norm for $t \in [0, T]$.*

**Assumption A.2** ( Numerical Stability). *The temporal discretization adheres to a stability constraint of the form:*

$$\Delta t \leq C_{stab} h^\beta, \tag{A.1}$$

*where $h$ is the spatial resolution, $\beta > 0$ is an exponent determined by the highest-order spatial derivative in the PDE, and $C_{stab} > 0$ is a scheme-dependent constant derived from stability analysis (e.g., CFL condition).*

**Assumption A.3** (Small Perturbation). *The perturbations $\epsilon$ are small, such that:*

- *Validity of Linearization: Perturbations $\epsilon$ satisfy $|\epsilon| \ll 1$, ensuring residual terms of order $\mathcal{O}(\epsilon^2)$ are negligible in Taylor expansions of relevant operators.*

- *Lipschitz Bound: There exists $L > 0$ such that the Jacobian $J(u) = \partial \mathcal{N}/\partial u$ satisfies:*

$$\|J(u)\|_{H^s} \leq L \quad \forall u \in H^s(\mathbb{X}), \tag{A.2}$$

*and perturbations $\epsilon$ are sufficiently small to ensure the perturbed solutions $u + \epsilon \delta u$ remain in a neighborhood where this bound holds uniformly.*

## B  Proofs

### B.1  Proof of Proposition 4.1

Consider an explicit Euler time-stepping scheme for clarity. The numerical solution evolves as:

$$u^{n+1} = \mathcal{T}_t[u^{n+1}, \mathcal{N}(u^n)] = u^n + \Delta t \mathcal{N}(u^n). \tag{B.1}$$

Based on eq. (7), the perturbed solution, subject to a solution perturbation $\epsilon_u^n$ at step $n$ and a source perturbation $\epsilon_s^n$ during the step, evolves as:

$$(u^{n+1} + \delta u^{n+1}) = (u^n + \epsilon_u^n) + \Delta t(\mathcal{N}(u^n + \epsilon_u^n) + \epsilon_s^n), \tag{B.2}$$

and by rearrangement

$$\frac{(u^{n+1} + \delta u^{n+1}) - (u^n + \epsilon_u^n)}{\Delta t} = \mathcal{N}(u^n + \epsilon_u^n) + \epsilon_s^n. \tag{B.3}$$

Subtracting the exact update from the perturbed update yields:

$$\frac{\delta u^{n+1} - \epsilon_u^n}{\Delta t} = \mathcal{N}(u^n + \epsilon_u^n) - \mathcal{N}(u^n) + \epsilon_s^n. \tag{B.4}$$

By assumption A.3, perturbations are small, allowing a first-order Taylor expansion of $\mathcal{N}(\cdot)$ around $u^n$:

$$\mathcal{N}(u^n + \epsilon_u^n) \approx \mathcal{N}(u^n) + J(u^n)\epsilon_u^n, \tag{B.5}$$

where $J(u^n) = \frac{\partial \mathcal{N}}{\partial u}\big|_{u^n}$ is the Jacobian of $\mathcal{N}$ evaluated at $u^n$. Substituting this into the error equation gives:

$$\frac{\delta u^{n+1} - \epsilon_u^n}{\Delta t} \approx J(u^n)\epsilon_u^n + \epsilon_s^n. \tag{B.6}$$

Rearranging for $\delta u^{n+1}$:

$$\delta u^{n+1} \approx \epsilon_u^n + \Delta t J(u^n)\epsilon_u^n + \Delta t \epsilon_s^n = (I + \Delta t J(u^n))\epsilon_u^n + \Delta t \epsilon_s^n. \tag{B.7}$$

Defining the error amplification matrix $G(u^n) := I + \Delta t J(u^n)$, we obtain eq. (8):

$$\delta u^{n+1} = G(u^n)\epsilon_u^n + \Delta t \epsilon_s^n.$$

This completes the proof.

## B.2  Proof of Lemma 4.1

We prove by induction on $k$. For $k = 1$, the claim is exactly eq. (8).

$$\delta u^{n+1} = G(u^n)\epsilon_u^n + \Delta t \epsilon_s^n, \tag{B.8}$$

Then for step $n + 2$ :

$$\delta u^{n+2} = \underbrace{G(u^{n+1})G(u^n)\epsilon_u^n + G(u^{n+1})\Delta t \epsilon_s^n}_{\text{From } n+1} + \underbrace{G(u^{n+1})\epsilon_u^{n+1} + \Delta t \epsilon_s^{n+1}}_{\text{From } n+2}. \tag{B.9}$$

Thus, for $k$ steps, the global error $\delta u^{n+k}$ is a sum over all perturbations introduced at each step $m$, amplified by the product of $G(u)$ matrices from subsequent steps:

$$\delta u^{n+k} = \sum_{m=0}^{k-1} \left( \prod_{i=m+1}^{k-1} G(u^{n+i}) \right) \left[ G(u^{n+m})\epsilon_u^m + \Delta t \epsilon_s^m \right]. \tag{B.10}$$

## B.3  Proof of Proposition 4.2

Based on assumption A.3, we have eq. (A.2), the norm of the error amplification matrix $G(u) = I + \Delta t J(u)$ is bounded:

$$\|G(u)\| = \|I + \Delta t J(u)\| \le \|I\| + \Delta t \|J(u)\| \le 1 + \Delta t L. \tag{B.11}$$

The product of amplification matrices is thus bounded by:

$$\left\| \prod_{i=m+1}^{k-1} G(u^{n+i}) \right\| \le \prod_{i=m+1}^{k-1} \|G(u^{n+i})\| \le (1 + \Delta t L)^{k-1-(m+1)+1} = (1 + \Delta t L)^{k-m-1}. \tag{B.12}$$

Consider the numerator of $R_k$ (contribution from $\epsilon_u$ only, i.e., $\epsilon_s^{(m)} = 0$ for all $m$):

$$\|\delta u^{n+k}\|_{\epsilon_s=0} = \left\| \sum_{m=0}^{k-1} \left( \prod_{i=m+1}^{k-1} G(u^{n+i}) \right) G(u^{n+m})\epsilon_u^{(m)} \right\| \tag{B.13}$$

$$\le \sum_{m=0}^{k-1} \left\| \prod_{i=m+1}^{k-1} G(u^{n+i}) \right\| \|G(u^{n+m})\| \|\epsilon_u^{(m)}\| \tag{B.14}$$

$$\le \sum_{m=0}^{k-1} (1 + \Delta t L)^{k-m-1}(1 + \Delta t L)\epsilon \tag{B.15}$$

$$= \epsilon \sum_{m=0}^{k-1} (1 + \Delta t L)^{k-m}. \tag{B.16}$$

Consider the denominator of $R_k$ (contribution from $\epsilon_s$ only, i.e., $\epsilon_u^{(m)} = 0$ for all $m$):

$$\|\delta u^{n+k}\|_{\epsilon_u=0} = \left\| \sum_{m=0}^{k-1} \left( \prod_{i=m+1}^{k-1} G(u^{n+i}) \right) \Delta t \epsilon_s^{(m)} \right\| \tag{B.17}$$

$$\leq \sum_{m=0}^{k-1} \left\| \prod_{i=m+1}^{k-1} G(u^{n+i}) \right\| \Delta t \|\epsilon_s^{(m)}\| \tag{B.18}$$

$$\leq \sum_{m=0}^{k-1} (1 + \Delta t L)^{k-m-1} \Delta t \epsilon \tag{B.19}$$

$$= \Delta t \epsilon \sum_{m=0}^{k-1} (1 + \Delta t L)^{k-m-1}. \tag{B.20}$$

The ratio $R_k$ is then approximately:

$$R_k = \frac{\|\delta u^{n+k}\|_{\epsilon_s=0}}{\|\delta u^{n+k}\|_{\epsilon_u=0}} \approx \frac{\epsilon \sum_{m=0}^{k-1} (1 + \Delta t L)^{k-m}}{\Delta t \epsilon \sum_{m=0}^{k-1} (1 + \Delta t L)^{k-m-1}}. \tag{B.21}$$

Both the numerator and the denominator are geometric series.

For the numerator, set $r = 1 + \Delta t L$, $j = k - m$, and simplify via the sum of a geometric series:

$$\sum_{m=0}^{k-1} r^{k-m} = \sum_{j=1}^{k} r^j = r \frac{r^k - 1}{r - 1}$$

Similarly, for the denominator, let $j = k - m - 1$, we can get:

$$\sum_{m=0}^{k-1} r^{k-m-1} = \sum_{j=0}^{k-1} r^j = \frac{r^k - 1}{r - 1}$$

Thus, we can simplify $R_k$ to:

$$R_k = \frac{\epsilon \sum_{m=0}^{k-1} r^{k-m}}{\Delta t \epsilon \sum_{m=0}^{k-1} r^{k-m-1}} = \frac{r \frac{r^k - 1}{r-1}}{\Delta t \frac{r^k - 1}{r-1}} = \frac{r}{\Delta t} = \frac{1 + \Delta t L}{\Delta t}. \tag{B.22}$$

This completes the proof.

It is worth noting that we have used an upper-bound instead of the exact products of $G(u)$ to simplify the ratio $R_k$. It may overestimate the true ratio, but since the matrix products $\left( \prod_{i=m+1}^{k-1} G(u^{n+i}) \right)$ appear in both the numerator and the denominator, applying upper bounds to both and taking their ratio largely preserves the order of magnitude. This simplification transforms a complex expression into a tractable form, enabling clear focus on the dominant scaling behavior of $R_k$, which is critical for perturbation and stability analysis. Numerical studies of the perturbation growth in chaotic and non-chaotic systems are shown to comply with this theoretical result in appendix C.

### B.4 Proof of Proposition 4.3

Consider the maximum Lyapunov exponent $\lambda_{\max}$ defined in eq. (12),

$$\lambda_{\max} = \limsup_{t \to \infty} \frac{1}{t} \ln \frac{\|\delta u(t)\|}{\|\delta u(0)\|} \tag{B.23}$$

Based on current PDE setup, $\delta u(t)$ satisfies the linearized equation:

$$\partial_t(\delta u) = J(u)\delta u, \quad J(u) = \frac{\partial \mathcal{N}}{\partial u}. \tag{B.24}$$

**Lipschitz constant** $L$    Typically, $L$ is defined via the nonlinear operator $\mathcal{N}$:

$$\|\mathcal{N}(u) - \mathcal{N}(v)\|_{H^s} \leq L\|u - v\|_{H^s}. \tag{B.25}$$

Since $\mathcal{N}$ is differentiable, apply the mean value theorem:

$$\|\mathcal{N}(u) - \mathcal{N}(v)\|_{H^s} \leq \sup_w \|J(w)\|_{H^s}\|u - v\|_{H^s}, \tag{B.26}$$

Then we define $L$ as the supremum of the operator norm of the Jacobian,

$$L = \sup_u \|J(u)\|_{H^s}, \quad \text{such that}\|J(u)\|_{H^s} \leq L \tag{B.27}$$

Using Grönwall's inequality on $\partial_t(\delta u) = J(u)\delta u$:

$$\|\delta u(t)\| \leq \|\delta u(0)\|e^{\int_0^t \|J(u(s))\|_{H^s}\,ds}. \tag{B.28}$$

Taking the logarithm and dividing by $t$:

$$\frac{1}{t}\ln\frac{\|\delta u(t)\|}{\|\delta u(0)\|} \leq \frac{1}{t}\int_0^t \|J(u(s))\|_{H^s}\,ds, \tag{B.29}$$

since $\|J(u(s))\|_{H^s} \leq L$, thus $\int_0^t \|J(u(s))\|_{H^s}\,ds \leq \int_0^t L\,ds = Lt$. Then:

$$\frac{1}{t}\ln\frac{\|\delta u(t)\|}{\|\delta u(0)\|} \leq L. \tag{B.30}$$

Since for every finite $t$ the quantity $\frac{1}{t}\ln\frac{\|\delta u(t)\|}{\|\delta u(0)\|}$ is bounded above by $L$, its $\limsup$ as $t \to \infty$ also cannot exceed $L$. Thus

$$\lambda_{\max} = \limsup_{t\to\infty}\frac{1}{t}\ln\frac{\|\delta u(t)\|}{\|\delta u(0)\|} \leq L. \tag{B.31}$$

The assertion that $\lambda_{\max} > 0$ for chaotic systems is a standard criterion for chaos, reflecting exponential divergence of trajectories. Combined with eq. (B.31), this gives:

$$L \geq \lambda_{\max} > 0. \tag{B.32}$$

This completes the proof. Meanwhile, the above proofs are for $\mathcal{N}$ is differentiable, which is valid for the current setup. As for non-differentiable functions, there exists a similar relation between $L$ and $\lambda_{\max}$; more details can be found in the literature [49]. The next section shows how the insights from our derivation above carry over to real-world experiments with non-linear PDEs.

## C    Numerical studies of the error dominance ratio

To investigate the sensitivity of different correction methods to perturbations, we introduce Gaussian noise as a perturbation at each time step via two distinct mechanisms: (1) **direct injection**, where the noise is directly added to the solution, mimicking existing neural correction methods like SITL. (2) **indirect injection**, where noise influences the solution indirectly, as employed in our proposed INC framework. The injected noise is sampled from a zero-mean Gaussian distribution, $\eta \sim \mathcal{N}(0, \epsilon^2 \cdot I)$, where $\epsilon$ controls the noise strength. To comprehensively study the impact of perturbations under varying dynamical regimes, we evaluate both a stable system (1D Burgers equation) and a chaotic system (1D Kuramoto–Sivashinsky equation). For Burgers, the perturbation magnitude is sampled in $\{10^{-4}, 10^{-2}, 1\}$, while for KS $\epsilon \in \{10^{-4}, 10^{-2}, 10^{-1}\}$.

The experimental results clearly demonstrate that solutions subjected to direct perturbation are significantly more sensitive compared to those influenced indirectly. In the Burgers scenario, injecting perturbations directly leads to rapid error accumulation. At high noise levels ($\epsilon = 1$), the rollout becomes numerically unstable, diverging almost immediately. Indirect perturbation, by contrast, maintains stable dynamics and low error accumulation across all tested values of $\epsilon$, as detailed in fig. 7. For KS, with its chaotic nature, perturbations grow exponentially over time. As a result, the MSE naturally increases along the rollout horizon, regardless of the perturbation magnitude and injection methods. Nevertheless, how the perturbations are introduced has a substantial impact on

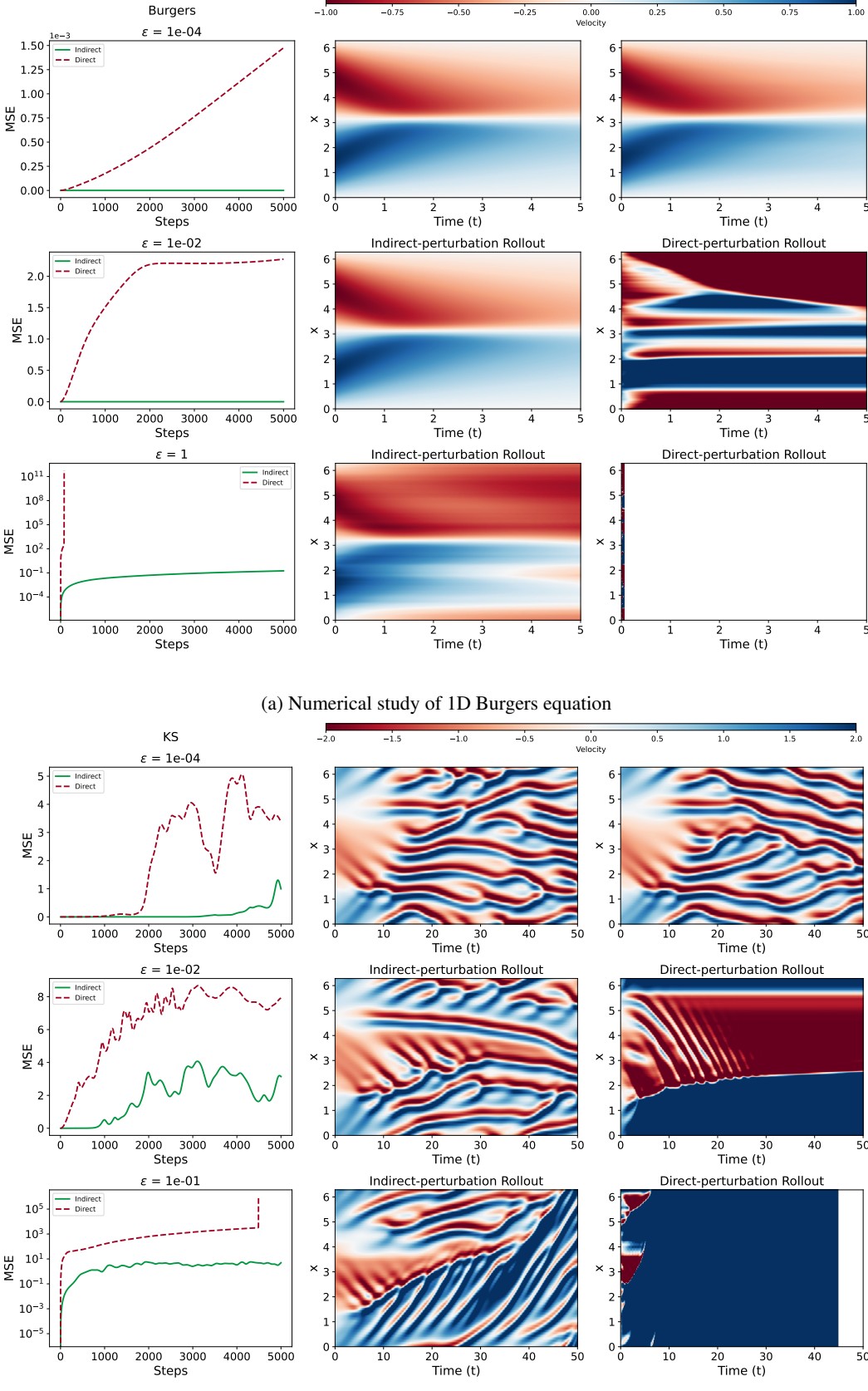

(a) Numerical study of 1D Burgers equation

(b) Numerical study of 1D KS equation

Figure 7: Numerical study of noise injected directly/indirectly, with indirect injection being more stable to noise

the rate and severity of this divergence. When perturbations are injected directly into the solution at each time step, the resulting trajectories exhibit rapid error amplification, with the MSE growing quickly and the rollouts becoming unstable or even collapsing under moderate to large perturbations. Conversely, when the perturbation is injected indirectly, the error growth is noticeably slower, and the rollout remains more coherent over longer timescales. This damping effect indicates that, compared with injecting perturbations directly, injecting indirectly can suppress the exponential sensitivity of the chaotic system, leading to improved numerical stability and more reliable long-term forecasts, even under significant noise levels. These empirical observations are consistent with and further support the theoretical analysis developed in this work.

## D   Details and schematic illustration

In this section, we provide more details about our Indirect Neural Correction (INC) method. The autoregressive rollout process for INC is illustrated in fig. 8. At each timestep $n$, the solution $u^n$ is updated through a temporal scheme $\mathcal{T}$, which integrates the physical dynamics governed by spatial derivatives $\mathcal{N}$ combined with a neural correction $\mathcal{G}_\theta(u^n)$. The neural corrector provides indirect corrections by modifying the PDE's right-hand side, influencing the system's evolution at every step. This correction strategy continues iteratively, performing $k$ sequential autoregressive rollouts. As depicted, the neural corrector output is integrated directly within each time-step operator, rather than being applied post hoc, fundamentally affecting the system's stability and long-term error propagation. The dashed lines indicate the feedback mechanism from the neural corrector to the spatial derivative calculation, highlighting the indirect nature of this correction and its role in stabilizing the numerical integration across multiple time steps.

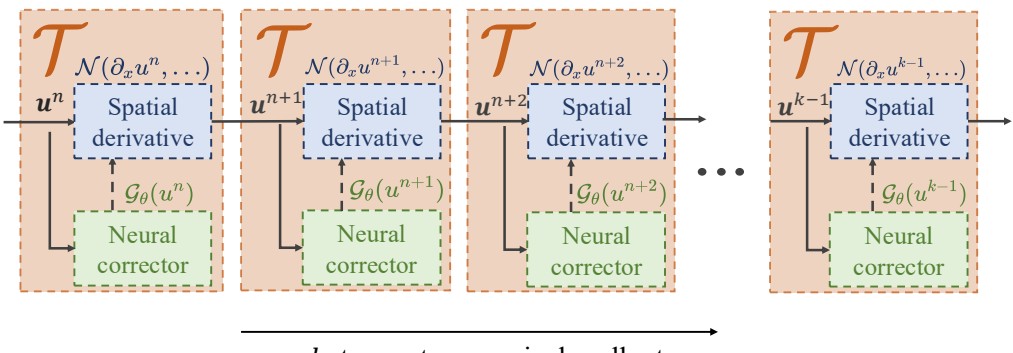

$k$ steps autoregressively rollout

Figure 8: Schematic illustration of our Indirect Neural Correction (INC) rollout procedure. At each timestep $n$, the solution $u^n$ is advanced through the temporal integrator $\mathcal{T}$, incorporating the physical dynamics governed by spatial derivatives $\mathcal{N}$ and indirect neural corrections $\mathcal{G}_\theta(u^n)$. This autoregressive approach is repeated iteratively for $k$ steps, highlighting how INC influences each integration step.

## E   Weighted Essentially Non-Oscillatory (WENO) based finite-difference method for Burgers

The following three sections provide details of the differentiable solvers that were used for the experiments in our paper. Details of the experiments will be provided afterwards.

The viscous Burgers equation represents a fundamental model combining nonlinear advection with diffusive effects:

$$\frac{\partial u}{\partial t} + \frac{\partial}{\partial x}\left(\frac{u^2}{2}\right) = \nu \frac{\partial^2 u}{\partial x^2}, \tag{E.1}$$

where $u(x,t)$ denotes the velocity field and $\nu$ the kinematic viscosity. This PDE system presents numerical challenges due to the shock formation while requiring accurate resolution.

## E.1 Weighted Essentially Non-Oscillatory (WENO) Scheme

The fifth-order WENO (WENO5) scheme, originally developed by Jiang and Shu [1, 50], provides an effective framework for resolving both smooth structures and discontinuities. Building upon the Essentially Non-Oscillatory (ENO) concept, WENO improves reconstruction accuracy through convex combination of multiple candidate stencils, weighted by local smoothness measures.

## E.2 Lax-Friedrichs Flux Splitting

To facilitate stable upwind reconstruction in characteristic space, we employ the local Lax-Friedrichs flux splitting technique [51]. This approach decomposes the nonlinear flux $f(u) = u^2/2$ into positively and negatively propagating components:

$$f^{\pm}(u) = \frac{1}{2}\left(f(u) \pm \alpha u\right), \tag{E.2}$$

where $\alpha = \max|u|$ represents the maximum characteristic speed over the computational domain. The splitting parameter $\alpha$ ensures the numerical flux Jacobians satisfy:

$$\frac{\partial f^+}{\partial u} \geq 0, \quad \frac{\partial f^-}{\partial u} \leq 0, \tag{E.3}$$

guaranteeing the upwind property for each component.

## E.3 Weighted Essentially Non-Oscillatory (WENO) Reconstruction Procedure

The fifth-order WENO (WENO5) algorithm constructs interface flux values through the following systematic process:

**Stencil Configuration**    For each interface $x_{i+1/2}$, three candidate stencils are considered for both flux components:

- Positive fluxes ($f^+$): Upwind-biased stencils

$$S_0^+ = \{x_{i-2}, x_{i-1}, x_i\}$$
$$S_1^+ = \{x_{i-1}, x_i, x_{i+1}\}$$
$$S_2^+ = \{x_i, x_{i+1}, x_{i+2}\}$$

- Negative fluxes ($f^-$): Downwind-biased stencils

$$S_0^- = \{x_{i+2}, x_{i+1}, x_i\}$$
$$S_1^- = \{x_{i+1}, x_i, x_{i-1}\}$$
$$S_2^- = \{x_i, x_{i-1}, x_{i-2}\}$$

This dual stencil arrangement ensures proper upwinding directionality for each flux component while maintaining the formal five-point spatial support.

**Polynomial Reconstruction**    Each 3-point stencil generates a quadratic polynomial $p_k(x)$ approximating the flux function. Through Taylor series expansion about $x_{i+1/2}$, we derive the optimal linear weights $\gamma_k$ that would yield fifth-order accuracy if applied uniformly in smooth regions. For the positive flux component, the interface extrapolations yield:

$$f_{i+1/2}^{(0)+} = \frac{1}{3}f_{i-2} - \frac{7}{6}f_{i-1} + \frac{11}{6}f_i \tag{E.4}$$

$$f_{i+1/2}^{(1)+} = -\frac{1}{6}f_{i-1} + \frac{5}{6}f_i + \frac{1}{3}f_{i+1} \tag{E.5}$$

$$f_{i+1/2}^{(2)+} = \frac{1}{3}f_i + \frac{5}{6}f_{i+1} - \frac{1}{6}f_{i+2} \tag{E.6}$$

Negative flux components employ symmetrized coefficients through index reflection about $x_{i+1/2}$.

**Smoothness Indicators**  Oscillation detection is quantified through the Sobolev-type smoothness measure:

$$\beta_k = \sum_{m=1}^{2} \Delta x^{2m-1} \int_{x_{i-1/2}}^{x_{i+1/2}} \left( \frac{d^m p_k}{dx^m} \right)^2 dx, \tag{E.7}$$

which penalizes high-order derivatives to detect nonsmooth features. Expanding for each stencil:

$$\beta_0 = \frac{13}{12}(f_{i-2} - 2f_{i-1} + f_i)^2 + \frac{1}{4}(f_{i-2} - 4f_{i-1} + 3f_i)^2$$
$$\beta_1 = \frac{13}{12}(f_{i-1} - 2f_i + f_{i+1})^2 + \frac{1}{4}(f_{i-1} - f_{i+1})^2$$
$$\beta_2 = \frac{13}{12}(f_i - 2f_{i+1} + f_{i+2})^2 + \frac{1}{4}(3f_i - 4f_{i+1} + f_{i+2})^2 \tag{E.8}$$

Smaller $\beta_k$ values indicate smoother stencils, preferentially weighted in the reconstruction.

**Adaptive Weighting Strategy**  Nonlinear weights combine stencil contributions while enforcing the ENO property:

$$\omega_k^{\pm} = \frac{\gamma_k}{(\epsilon + \beta_k^{\pm})^2}, \quad \epsilon = 10^{-12}, \tag{E.9}$$

where $\gamma = [0.1, 0.6, 0.3]$ are the optimal linear weights for smooth solutions. The regularization parameter $\epsilon$ prevents division by zero while maintaining weights near optimal values in smooth regions. Final weights are normalized as:

$$\omega_k^{\pm} \leftarrow \frac{\omega_k^{\pm}}{\sum_{m=0}^{2} \omega_m^{\pm}} \tag{E.10}$$

This weighting strategy automatically assigns dominant weight to the smoothest stencil while retaining high-order accuracy when all stencils are smooth.

**Flux Reconstruction**  The interface flux combines weighted contributions from all stencils:

$$\hat{f}_{i+1/2} = \sum_{k=0}^{2} \left( \omega_k^+ f_{i+1/2}^{(k)+} + \omega_k^- f_{i+1/2}^{(k)-} \right) \tag{E.11}$$

The complete WENO5 procedure achieves fifth-order accuracy in smooth regions while providing non-oscillatory shock transitions.

The diffusion term employs second-order central differences:

$$\left. \frac{\partial^2 u}{\partial x^2} \right|_i = \frac{u_{i+1} - 2u_i + u_{i-1}}{\Delta x^2} + \mathcal{O}(\Delta x^2) \tag{E.12}$$

### E.4   Temporal Integration

A forward Euler method is used to advance the solution:

$$u^{n+1} = u^n + \Delta t \left[ -\frac{\partial}{\partial x} \left( \frac{u^2}{2} \right)^n + \nu \left. \frac{\partial^2 u}{\partial x^2} \right|^n \right] \tag{E.13}$$

In order to ensure stability, adaptive time forward is supported with a constrained time step by the CFL condition:

$$\Delta t \leq \text{CFL} \cdot \min \left( \frac{\Delta x}{\max |u^n|} \right) \tag{E.14}$$

where CFL $\in (0, 1]$ is a user-specified parameter. The code implements adaptive sub-stepping to guarantee this condition at each iteration.

Periodic boundaries are enforced through index mapping:

$$u_{i+k} = u_{(i+k) \bmod N}, \quad k \in \{-2, -1, 0, 1, 2\} \tag{E.15}$$

### E.5 Validation Cases

To validate our solver, we consider two problems with known analytical solutions: a homogeneous case and a case with a quadratic source term. The former checks pure advection, while the latter verifies correct handling of source terms.

**Homogeneous Burgers Equation**   We solve the homogeneous Burgers equation
$$u_t + u\, u_x = 0, \qquad u(x,0) = h_0(x) = \sin\!\left(\tfrac{2\pi x}{L}\right), \tag{E.16}$$
on the periodic domain $x \in [0, L]$ with $L = 2$. The entropy solution can be written via the Hopf–Lax formula [52]:

$$u(x,t) = \arg\min_y \left\{ F(y) + \frac{(x-y)^2}{2t} \right\}, \tag{E.17}$$

where $F(y) = -\frac{1}{\pi}\cos(\pi y)$ is the antiderivative of the initial condition $u_0$.

Figure 9 compares the numerical and analytical solutions from $t = 0$ to $2\,\mathrm{s}$, demonstrating excellent agreement.

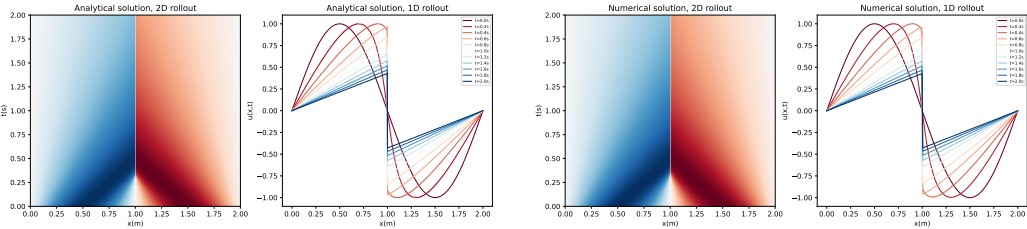

Figure 9: Analytical (left) vs. numerical (right) solutions of the homogeneous Burgers equation.

**Burgers Equation with a Quadratic Source Term**   Next, we test the solver on
$$u_t + u\, u_x = \beta\, u^2, \qquad u(x,0) = h_0(x), \tag{E.18}$$
again with periodic boundary conditions on $x \in [0, 1]$. Using the method of characteristics [53], one obtains
$$u(x,t) = \frac{h_0(y)}{1 - \beta t h_0(y)}, \quad \text{where } y - \frac{\ln(1 - \beta t h_0(y))}{\beta} = x \tag{E.19}$$

For $\beta = -2$ and $h_0(x) = \sin(2\pi x)$ this simplifies to
$$u(x,t) = \frac{\sin(2\pi y)}{1 + 2t\sin(2\pi y)}, \quad \text{where } y + \frac{1}{2}\ln(1 + 2t\sin(2\pi y)) = x. \tag{E.20}$$

Figure 10 shows that the numerical solution aligns well with the analytical one for $t \in [0, 0.15]\,\mathrm{s}$.

## F   Fourier Pseudo-Spectral Exponential Time Differencing Method for Kuramoto-Sivashinsky equation

### F.1   Governing Equations

The one-dimensional Kuramoto-Sivashinsky (KS) equation governs spatiotemporal pattern formation in dissipative systems:
$$\frac{\partial u}{\partial t} + u\frac{\partial u}{\partial x} + \frac{\partial^2 u}{\partial x^2} + \frac{\partial^4 u}{\partial x^4} = 0, \tag{F.1}$$
where $u(x,t)$ represents the scalar field of interest. The equation combines:

- Nonlinear advection through the $u\partial u/\partial x$ term
- Energy production/dissipation via the $\partial^2 u/\partial x^2$ operator
- High-wavenumber stabilization from the $\partial^4 u/\partial x^4$ hyperviscosity

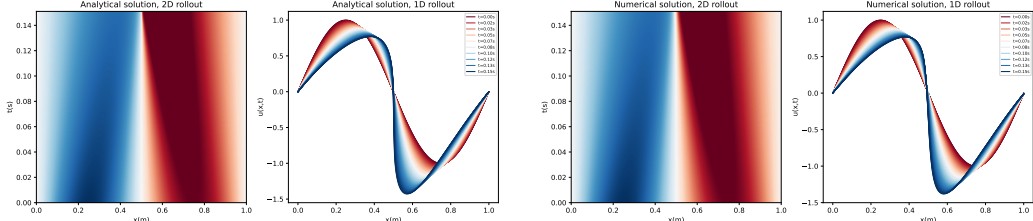

Figure 10: Analytical (left) vs. numerical (right) solutions of the Burgers equation with quadratic source term.

## F.2 Spectral Discretization

We employ a pseudospectral method with periodic boundary conditions on domain $x \in [0, L]$. The solution is discretized over $N$ collocation points and transformed to Fourier space via:

$$\hat{u}(k_n, t) = \mathcal{F}\{u(x,t)\} = \sum_{j=0}^{N-1} u(x_j, t)e^{-ik_n x_j}, \tag{F.2}$$

where wavenumbers $k_n = \frac{2\pi n}{L}$ for $n \in [-N/2 + 1, N/2]$.

Spatial derivatives become algebraic operations in Fourier space:

$$\mathcal{F}\left\{\frac{\partial^m u}{\partial x^m}\right\} = (ik)^m \hat{u}(k, t). \tag{F.3}$$

The linear operator for KS becomes:

$$\mathcal{L} = -((ik)^2 + (ik)^4). \tag{F.4}$$

The advection term transforms via convolution theorem:

$$\mathcal{F}\left\{u\frac{\partial u}{\partial x}\right\} = -\frac{ik}{2}\mathcal{F}\{u^2\}, \tag{F.5}$$

The semi-discrete Fourier-transformed KS equation becomes:

$$\frac{d\hat{u}}{dt} = \mathcal{L}\hat{u} + \mathcal{N}(\hat{u}), \tag{F.6}$$

where $\mathcal{N}(\hat{u}) = -\frac{ik}{2}\mathcal{F}\{u^2\}$ contains nonlinear terms.

## F.3 Exponential Time Differencing

Using the variation-of-constants formula, the exact solution evolves as:

$$\hat{u}(t + \Delta t) = e^{\mathcal{L}\Delta t}\hat{u}(t) + \int_0^{\Delta t} e^{\mathcal{L}(\Delta t - \tau)}\mathcal{N}(\hat{u}(t + \tau))d\tau. \tag{F.7}$$

For the non-linear part, there are different scheme to approximate,

Approximating the nonlinear term as constant over $\tau \in [0, \Delta t]$:

$$\hat{u}^{n+1} = e^{\mathcal{L}\Delta t}\hat{u}^n + \mathcal{L}^{-1}(e^{\mathcal{L}\Delta t} - I)\mathcal{N}(\hat{u}^n). \tag{F.8}$$

We also implement a higher order of temporal scheme, second-order Runge-Kutta scheme (ETDRK2) to improves temporal accuracy:

$$\hat{u}_* = e^{\mathcal{L}\Delta t}\hat{u}^n + \mathcal{L}^{-1}(e^{\mathcal{L}\Delta t} - I)\mathcal{N}(\hat{u}^n) \tag{F.9}$$

$$\hat{u}^{n+1} = \hat{u}_* + \frac{e^{\mathcal{L}\Delta t} - I - \mathcal{L}\Delta t}{\mathcal{L}^2\Delta t}(\mathcal{N}(\hat{u}_*) - \mathcal{N}(\hat{u}^n)). \tag{F.10}$$

### F.4 Source Term Incorporation

For forced systems $\partial_t u = \cdots + S(x, t)$, the semi-discrete equation becomes:

$$\frac{d\hat{u}}{dt} = \mathcal{L}\hat{u} + \mathcal{N}(\hat{u}) + \hat{S}. \tag{F.11}$$

The ETD framework naturally extends through modified integration:

$$\hat{u}^{n+1} = e^{\mathcal{L}\Delta t}\hat{u}^n + \int_0^{\Delta t} e^{\mathcal{L}(\Delta t - \tau)}(\mathcal{N}(\hat{u}^n + \tau)) + \hat{S})d\tau, \tag{F.12}$$

with source terms treated analogously to nonlinear terms in temporal discretization.

## G Pressure Implicit with Splitting of Operators method (PISO) for incompressible Navier–Stokes equation

The Pressure Implicit with Splitting of Operators (PISO) algorithm is a widely used predictor-corrector method in computational fluid dynamics [3]. It can be written as:

### G.1 Predictor Step

The velocity predictor equation is given by:

$$\frac{1}{\Delta t}\mathbf{u}^* + \nabla \cdot (\mathbf{u}^n\mathbf{u}^*) - \nu\nabla^2\mathbf{u}^* = \frac{1}{\Delta t}\mathbf{u}^n - \nabla p + \mathbf{S}^n, \tag{G.1}$$

where $\mathbf{u}^*$ is the predicted velocity field, $\nu$ is the kinematic viscosity, and $\mathbf{S}^n$ represents the source term at time step $n$.

Rewriting this equation in matrix form:

$$C\mathbf{u}^* = \frac{1}{\Delta t}\mathbf{u}^n - \nabla p + \mathbf{S}^n. \tag{G.2}$$

### G.2 Pressure Correction Step

For the corrector step, the matrix $C$ is decomposed into its diagonal component $A$ and off-diagonal component $H$. Defining:

$$h = -H\mathbf{u}^* + \frac{\mathbf{u}^n}{\Delta t}, \tag{G.3}$$

the predicted velocity $\mathbf{u}^*$ can be expressed as:

$$\mathbf{u}^* = A^{-1}h - A^{-1}\nabla p + A^{-1}\mathbf{S}^n. \tag{G.4}$$

Applying the divergence-free constraint $\nabla \cdot \mathbf{u}^{**} = 0$, we obtain:

$$\nabla \cdot (A^{-1}h - A^{-1}\nabla p^* + A^{-1}\mathbf{S}^n) = 0. \tag{G.5}$$

Defining the intermediate velocity correction term:

$$\mathbf{h} = A^{-1}(h + \mathbf{S}^n), \tag{G.6}$$

we arrive at the pressure Poisson equation:

$$\nabla \cdot (A^{-1}\nabla p^*) = \nabla^2(A^{-1}p^*) = \nabla \cdot \mathbf{h}. \tag{G.7}$$

### G.3 Velocity Correction

Once the pressure $p^*$ is solved, the velocity field is corrected to ensure divergence-free conditions:

$$\mathbf{u}^{**} = \mathbf{h} - A^{-1}\nabla p^*. \tag{G.8}$$

The corrected velocity $\mathbf{u}^{**}$ is then used in the next iteration of the correction loop, repeating the pressure correction step to further improve accuracy. Typically, this correction process is performed twice to enhance solution stability.

Table 1: Summary of numerical solvers for different cases. Temporal schemes and spatial discretizations are validated for each case.

| Case | Dimension | Temporal Scheme | Spatial Discretization |
|---|---|---|---|
| Kuramoto–Sivashinsky (KS1/KS2) | 1D | ETDRK | Pseudo-Spectral |
| Burgers' Equation (Burgers) | 1D | forward Euler | WENO-5 (Finite Difference) |
| Kármán Vortex Street (Karman) | 2D | Semi-implicit (PISO) | Finite Volume Method |
| Backward–Facing Step (BFS) | 2D | Semi-implicit (PISO) | Finite Volume Method |
| Turbulent Channel Flow (TCF) | 3D | Semi-implicit (PISO) | Finite Volume Method |

# H    Training details

Our experiments employ three different solvers with substantially different numerical schemes. An overview is given in table 1.

## H.1    Experiment 1 & 3: Kuramoto–Sivashinsky equation

**Numerical setup**    The Kuramoto–Sivashinsky (KS) equation serves as a prototypical example of spatiotemporal chaos. In this study, we consider two experimental setups. The first, denoted as KS1, operates at a fine temporal resolution with $\Delta t = 10^{-2}$ and extends for 5000 steps, as detailed in section 6.1, to assess model performance under extreme multi-step rollout conditions. The second, referred to as KS2, employs an extremely coarse time step of $\Delta t = 0.5$, which makes the simulation blow up typically within 28 steps. This setup is designed to test the capability of improving the numerical stability of the hybrid solver in a numerically unstable setting, as discussed in section 6.2. For both KS1 and KS2, the total simulation time is held constant at 50s, and the same first-order Runge–Kutta (RK1) temporal integration scheme is used. This simulation time corresponds to approximately $4.5\, t_{c,KS}$, where the characteristic timescale is defined by the inverse of the maximum Lyapunov exponent [18]:

$$t_{c,KS} = \frac{1}{\lambda_{max}} = \frac{1}{0.0882} \approx 11.3\,s. \tag{H.1}$$

The domain size $L$ plays a central role in determining the chaotic dynamics of the KS system [18, 41, 42]. To evaluate generalization, we vary $L$ between training and testing: the training domain lengths are set to $L_{train} = 2\pi \cdot \{6.4, 7.2, 8.0, 8.8, 9.6, 10.4\}$, and the testing domain lengths are $L_{test} = 2\pi \cdot \{6.0, 8.4, 10.8\}$. The spatial resolution is fixed at 64 grid points for all experiments.

**Training setup**    INC, SITL and SITL* are trained together with the differentiable solver. I.e., they employ *solver-in-the-loop* training with a multi-step unrolled optimization strategy, each step being supervised by a high-fidelity reference state. The training data is generated from high-fidelity simulations and downsampled to a low resolution. During the training phase, the neural corrector is optimized to minimize the discrepancy between trajectories produced by the hybrid solver (the coarse numerical solver combined with the neural corrector) and the downsampled high-fidelity solutions. The loss function is an $L^2$ loss computed over multi-step autoregressive rollouts. This multi-step training is conducted in an end-to-end differentiable framework (all solvers are implemented in PyTorch and CUDA), allowing gradients to backpropagate through both the neural corrector and the numerical solver steps for the whole rollout. This training paradigm ensures that the neural corrector learns corrections optimized explicitly for long-term stability and accuracy, and accounts for the solver's intrinsic numerical dynamics. In our work, the choice of unrolling length is guided by the characteristic timescale $t_c$ of the dynamical systems. For each case, $t_c$ is derived from system-specific properties, like the inverse of the maximum Lyapunov exponent for the Kuramoto–Sivashinsky equation. More details about $t_c$ for each case are detailed in the following paragraph. After calculation of $t_c$ and setting the time step as $\Delta t$, we compute the number of steps per characteristic timescale as $N = \frac{t_c}{\Delta t}$. We then set the unrolling length during training to approximately $0.04 \cdot N$. During inference, we significantly extend this horizon by a factor of about 100, typically to $4\cdot N$ – $6\cdot N$, to rigorously evaluate long-term stability and accuracy. We believe that anchoring both training and inference lengths to the system's intrinsic timescale results in a more physically grounded and robust learning process. Similar to the training setup, the hybrid solver couples neural networks with a low-resolution solver also in the inference phase.

We evaluate four distinct network architectures, comprising both convolution-based and operator-based models. Specifically, we deploy ResNet and UNet as convolutional models, and Fourier Neural Operator (FNO) and DeepONet as operator-learning approaches. For neural correctors (INC, SITL, SITL*), the input consists of two channels: the solution $u$ and the corresponding positional grid. For recurrent architectures such as RNN, we follow the setup of Li et al.[4], using 11 input channels (10 historical time steps plus 1 positional grid).

*ResNet* [54] is implemented as a 1D architecture composed of six residual blocks, each with 32 feature channels. Circular padding is applied to Conv1D layers (kernel size 5), and identity mappings are used as residual connections. ReLU is adopted as the activation function throughout. A final $1 \times 1$ Conv1D layer maps features to a single output channel. The total number of trainable parameters is 62.2k for correctors and 63.6k for RNN.

*UNet* [55] adopts a symmetric encoder–decoder design, initialized with 32 base features. The network includes circularly padded Conv1D layers (kernel size 5) with ReLU activations, max-pooling (kernel size 2) in the encoder, transposed convolutions (kernel size 2) in the decoder, and skip connections to preserve high-resolution features. A final $1 \times 1$ Conv1D produces the output. The parameter count is 257.1k for correctors and 258.5k for RNN.

*FNO* [4] begins with a linear layer that maps the input into a feature space with 64 channels, followed by four Fourier layers, each multiplying 16 modes in the frequency domain, plus $1 \times 1$ Conv1d residual branches and GeLU activations. Total trainable parameters are 549.6k/550.1k for correctors/RNN.

*DeepONet* [9] utilizes a branch–trunk architecture. The branch net processes inputs at 100 sensor points using four fully connected layers with 128 neurons and Tanh activations. The trunk net mirrors this structure, with transfer of the input to a Fourier basis function (4 modes). Outputs from the branch and trunk networks are combined via a dot product and a learnable bias term. The total parameter count is 113.4k for correctors and 228.6k for the RNN.

All models are trained using the Adam optimizer with a fixed learning rate of $10^{-4}$, an $\ell_2$ regularization coefficient of $10^{-7}$, and a batch size of 64. Training is conducted for 100 epochs using a multi-step rollout strategy, with 100 time steps for KS1 and 10 time steps for KS2. The iterations are 42k in total. To prevent overfitting, 10% of the training data is set aside for validation. The model exhibiting the best validation loss is selected as the final checkpoint. A learning rate scheduler (ReduceLROnPlateau) is employed, which reduces the learning rate by a factor of 0.5 after 5 epochs without improvement in validation loss. The minimum learning rate is capped at $10^{-7}$. An early stopping criterion halts training if no improvement is observed for 15 consecutive epochs. The training time is approximately 2 hours for KS1 and 4 hours for KS2 on a single NVIDIA RTX 2080 Ti GPU. This variation arises from the differing rollout lengths used during training.

## H.2  Experiment 2: Burgers equation

**Numerical setup**  For Burgers, our objective is to evaluate how neural correctors perform under spatially downsampled settings with long-term rollout (4000 steps) and a fine temporal resolution ($\Delta t = 10^{-3}$). The total simulation time spans 4s, which corresponds to at least $4t_{c,BG}$, where $t_{c,BG}$ denotes the characteristic time at which the first pair of characteristics intersect and is determined numerically. Spatial downsampling is applied to the reference solution with a resolution of 512, producing coarser grids at resolutions 128, 64, 32, and 16—corresponding to downsampling factors of $4\times$, $8\times$, $16\times$ and $32\times$, respectively. To further evaluate the generalization capability of the neural correctors beyond random initial condition variations (as discussed in section 5), we introduce a viscosity shift between training and testing. Specifically, the viscosity is set to $\nu_{train} = 0.2$ and $\nu_{test} = 0.25$.

**Training setup**  The neural network configurations mirror those used in the KS experiments. For all models, the architecture and training procedures remain identical, with one exception: for FNO at the lowest resolution (16 grid points), the number of retained Fourier modes is reduced from 16 to 8 to accommodate the smaller spatial domain. Herein, the unrolled step for training is 50. Under this modification, the number of trainable parameters for FNO becomes 287.4k for correctors and 288.0k for RNN. The training procedure also closely follows that of the KS setup, albeit with one adjustment. To account for the increased numerical sensitivity in Burgers simulations, the learning rate is slightly reduced to $10^{-5}$. All other hyperparameters—including batch size, optimizer (Adam),

regularization strategy, learning rate scheduling, and early stopping criteria—remain consistent with the KS training configuration. The training time ranges from approximately 10 to 18 hours on a single NVIDIA RTX 2080 Ti GPU. The variation is primarily due to differences in spatial resolution across cases and the use of an early stopping mechanism, which terminates training based on validation performance.

### H.3    Experiment 4: Karman

**Numerical setup**    2D Karman vortex shedding is a classic benchmark problem controlled by the incompressible Navier–Stokes equation. In this study, we focus on turbulent regimes characterized by Reynolds numbers in Re $\in \{500, 600\}$, using a rectangular obstacle placed centrally within the domain. The sharp corners of the obstacle induce localized disturbances that propagate across the domain, introducing numerical challenges due to their sensitivity to spatial resolution and stability. These disturbances can generate checkerboard artifacts when the flow is simulated under coarse discretization, as illustrated in fig. 3b for the No Model at 10s. In this study, we apply a $4\times$ downsampling in both spatial directions, yielding a coarse grid resolution of $67 \times 36$. Additionally, we set the CFL number to 2, which is relatively high compared to the typical stability criterion of CFL $< 1.0$, further challenging the numerical solver. To test the generalization of the corrector, we vary the obstacle height $y$, with training configurations sampled from $y_{\text{train}} \in \{1.0, 1.5\}$ and testing conducted at $y_{\text{test}} = 2.0$.

**Training setup**    For neural network architecture, this scenario used a 2D CNN with 7 layers and 16, 32, 64, 64, 64, 64, and 2 filters, respectively. The kernel sizes of the filters are $7^2, 5^2, 5^2, 3^2, 3^2, 1^2$, and $1^2$, respectively, for a total of 144.7k parameters. The stride is 1 in all layers, and we use ReLU as the activation function. Given the variation in Reynolds number and geometry across simulations, the input channel includes both parameters in addition to the 2D velocity fields, resulting in four input channels.

Training is conducted using the Adam optimizer with a learning rate of $10^{-4}$ with a fixed weight decay $10^{-1}$. Due to the increased numerical difficulty and sensitivity in 2D simulations, early-stage instability poses a significant challenge during training. To mitigate this issue, a staged multi-step rollout strategy is employed. The model is first trained with a rollout length of 2 steps for 24,000 iterations, followed by 8 steps for an additional 24,000 iterations. This progressive increase in rollout horizon is crucial for stabilizing the learning process during the initial phase of training. The total training time is approximately 13 hours on a single NVIDIA RTX 2080 Ti GPU.

### H.4    Experiment 5: Backward facing step

**Numerical setup**    The Backward-facing step (BFS) is a classic example of separated flow induced by an abrupt geometric expansion. The flow evolves spatially along the downstream region, requiring long-term accuracy and stability to consistently reproduce turbulent statistics [56, 57]. Furthermore, BFS is inherently sensitive to resolution due to grid-induced oscillations [58]. In our numerical setup, we apply a $4\times$ spatial and $20\times$ temporal downsampling. This setup results in a CFL number of CFL $= 4.0$, with the time step set to $\Delta t = 0.1$. The spatial resolution in the core domain—encompassing the region of vortex separation and reattachment, excluding the inlet and outlet, is set to $128 \times 32$. To validate the efficiency of INC, we compare it with two high-resolution simulations performed under numerically stable settings (CFL $< 0.8$), using resolutions of $192 \times 48$ and $256 \times 64$, donated as No Model$_{192}$ and No Model$_{256}$, respectively.

Two primary parameters strongly influence BFS flow behavior: the expansion ratio ($H/h$) and the Reynolds number (Re $= 2hU_b/\nu$), as they significantly affect critical metrics such as reattachment length [56, 57]. To further challenge the hybrid solver, we vary both the viscosity $\nu$ and the height $h$ (the height of the gap between the step and the top wall). During training, the parameters are sampled from $h_{\text{train}} \in \{0.85, 0.875\}$ and Re$_{\text{train}} \in \{1300, 1350\}$; for testing, they are set to $h_{\text{test}} = 1.0$ and Re$_{\text{test}} = 1400$. We simulate 1200 steps during testing, corresponding to a physical time of $t \cdot \frac{h}{U_b} = 4t_{\text{c,BFS}}$, where $U_b$ is the bulk velocity. The $t_{\text{c,BFS}}$ is defined as the time required for fluid to convect through the entire post-expansion region [59].

**Training setup**  The neural network architecture and training procedure employed for BFS follow the same configuration of Karman as described before. The total training time is about 16h on a single NVIDIA RTX 2080 Ti GPU.

### H.5   Experiment 6: Turbulent channel flow

**Numerical setup**  A turbulent channel flow (TCF) is a canonical wall-bounded shear flow governed by the incompressible Navier–Stokes equations. It is extensively employed to investigate the dynamics of near-wall turbulence and to evaluate the performance of turbulence modeling approaches. In this study, we consider the case of $\mathrm{Re}_\tau = 550$, utilizing a coarse spatial resolution of $64 \times 32 \times 32$. For SITL, SITL* and No Model, a CFL=0.8 is chosen to ensure numerical stability. In contrast, for INC, owing to its stabilizing properties, a $2\times$ larger time step is applied. This corresponds to a CFL number of 1.6, which is unstable for other models, yielding $\Delta t = 0.11$. To facilitate comparison, a high-resolution baseline case with $256 \times 128 \times 128$, referred to as No Model$_{256}$ is employed. The No Model$_{256}$ is under extreme numerical stability constraints with CFL=0.1. The distinction between training and testing lies in the number of forward steps. During training, the model is exposed to 12 multi-step rollouts, with a maximum warm-up period of 96 steps. Notably, the warm-up phase is used solely for forward simulation and does not involve gradient backpropagation. For evaluation, the models are tested over 1500 time steps, corresponding to $6\mathrm{t}_{c,\mathrm{TCF}}$ [60]. Here, the characteristic time scale $\mathrm{t}_{c,\mathrm{TCF}}$ is defined as $t \cdot \frac{u_\tau}{\delta}$, where $\delta$ is the ratio of the channel half-width and $u_\tau$ is the wall friction velocity. This timescale characterizes the evolution period of the largest energy-containing structures in the flow.

**Training setup**  For 3D TCF, we address a particularly challenging learning objective: training hybrid solvers to match turbulence statistics generated by a high-fidelity spectral solver at high resolution, rather than directly reproducing individual grid values. To ensure that the predicted flow fields accurately reflect the target turbulence statistics both globally and locally, the statistical loss is decomposed into two components: the averaged statistics loss during rollout training, and the frame-by-frame statistics loss, as described in detail below.

Consider a sequence of predicted velocity fields $u_i^n(x, y, z)$ at time steps $n = 0, 1, \ldots, N$, where index $i$ represents the velocity component (e.g., $i = 1, 2, 3$ for streamwise $x$, wall-normal $y$, and spanwise directions $z$). It's generated autoregressively from an initial state $\tilde{u}_i^0$ via a hybrid solver such that:

$$u_i^n = (\mathcal{T}|\mathcal{N}|\mathcal{G}_\theta)^n(\tilde{u}_i^0), \tag{H.2}$$

For each step $n$, we compute spatial averages over the homogeneous directions (streamwise $x$ and spanwise $z$):

$$\langle u_i \rangle^n(y) = \frac{1}{XZ} \sum_{x=0}^{X-1} \sum_{z=0}^{Z-1} u_i^n(x, y, z), \tag{H.3}$$

where $X$ and $Z$ are the grid resolutions in $x$ and $z$. The velocity fluctuation at step $n$ is:

$$u_i'^n(x, y, z) = u_i^n(x, y, z) - \langle u_i \rangle^n(y), \tag{H.4}$$

and the spatial Reynolds stress is:

$$\langle u_i' u_j' \rangle^n(y) = \frac{1}{XZ} \sum_{x=0}^{X-1} \sum_{z=0}^{Z-1} u_i'^n(x, y, z) u_j'^n(x, y, z). \tag{H.5}$$

The per-frame loss terms enforce statistical fidelity at each step $n$:

$$L_{u_i}^n = \frac{1}{Y} \sum_{y=0}^{Y-1} \mathcal{L}_2 \left( \langle u_i \rangle^n(y), \overline{\tilde{u}}_i(y) \right), \tag{H.6}$$

$$L_{u_{ij}'}^n = \frac{1}{Y} \sum_{y=0}^{Y-1} \mathcal{L}_2 \left( \langle u_i' u_j' \rangle^n(y), \overline{\tilde{u}_i' \tilde{u}_j'}(y) \right), \tag{H.7}$$

where $Y$ is the wall-normal resolution, and $\overline{\tilde{u}}_i(y)$ is the target time-averaged mean velocity from a high-fidelity spectral solver at high resolution[61].

For the averaged statistical losses over N steps, we define:

$$\overline{u_i}^{0:N}(y) = \frac{1}{N} \sum_{n=1}^{N} \langle u_i \rangle^n(y), \tag{H.8}$$

$$\overline{u_i' u_j'}^{0:N}(y) = \frac{1}{N} \sum_{n=1}^{N} \left( \frac{1}{XZ} \sum_{x=0}^{X-1} \sum_{z=0}^{Z-1} \left( u_i^n(x,y,z) - \overline{u_i}^{0:N}(y) \right) \left( u_j^n(x,y,z) - \overline{u_j}^{0:N}(y) \right) \right), \tag{H.9}$$

yielding:

$$L_{u_i}^{0:N} = \frac{1}{Y} \sum_{y=0}^{Y-1} \mathcal{L}_2 \left( \overline{u_i}^{0:N}(y), \widetilde{\overline{u}}_i(y) \right), \tag{H.10}$$

$$L_{u_{ij}'}^{0:N} = \frac{1}{Y} \sum_{y=0}^{Y-1} \mathcal{L}_2 \left( \overline{u_i' u_j'}^{0:N}(y), \widetilde{\overline{u_i' \widetilde{u}_j'}}(y) \right). \tag{H.11}$$

The total statistical loss combines these terms with weighting coefficients:

$$L_{\text{stats}} = \underbrace{\sum_{i=1}^{3} \lambda_{u_i} L_{u_i}^{0:N} + \sum_{i=1}^{3} \sum_{j=1}^{3} \lambda_{u_{ij}'} L_{u_{ij}'}^{0:N}}_{\text{mean stats loss}} + \underbrace{\sum_{n=0}^{N} \lambda_{\text{stats}}^n \left( \sum_{i=1}^{3} \lambda_{u_i} L_{u_i}^n + \sum_{i=1}^{3} \sum_{j=1}^{3} \lambda_{u_{ij}'} L_{u_{ij}'}^n \right)}_{\text{frame-by-frame stats loss}}, \tag{H.12}$$

where $\lambda_{u_i}$, $\lambda_{u_{ij}'}$, and $\lambda_{\text{stats}}^n$ are hyperparameters tuning the contribution of each term. Specifically, we set $\lambda_{u_1} = 1$ $\lambda_{u_2} = \lambda_{u_3} = 0.5$, $\lambda_{u_{ij}'} = 1.0$ and $\lambda_{\text{stats}}^n = 0.5$.

To quantitatively assess the accuracy of the hybrid solver in TCF case, we compute a statistical error metric that aggregates multiple normalized mean squared errors across key turbulence statistics. The error is defined as

$$L_{\text{MSE}_{\text{stats}}} = \sum_{q_i \in Q} \max |q_i^{\text{ref}}(y)| \frac{1}{Y} \sum_{y=0}^{Y-1} \mathcal{L}_2 \left( q_i(y), q_i^{\text{ref}}(y) \right) \Delta y \tag{H.13}$$

where $Q = \left\{ \overline{u_1}/u_\tau, \overline{u'u'}, \overline{v'v'}, \overline{w'w'}, \overline{u'v'} \right\}$ is the set of statistics, corresponding to mean streamwise velocity and key Reynolds stress components.

For this 3D case, we employ a convolution-based network as the neural corrector. The network architecture consists of successive convolutional layers with 8, 64, 64, 32, 16, 8, 4, and 3 channels, respectively. Each layer uses a kernel size of $3^3$, except for the last layer, which employs a kernel size of $1^3$. This gives 198.9k trainable parameters in total. ReLU activations are used for all but the last layer. The input to the network comprises four channels: the three components of the 3D velocity field and a normalized wall distance term defined as $1 - |\frac{y}{\delta}|$.

Training is performed using the Adam optimizer. The warm-up strategy is staged to facilitate stable learning. Initially, the network is trained for 6,000 iterations without any warm-up steps. This is followed by 20,000 iterations with warm-up steps randomly sampled from the range $[0, 12]$, 8,000 iterations with warm-up in $[0, 24]$, 20 iterations with warm-up in $[0, 48]$, and a final 20 iterations with warm-up in $[0, 96]$. To accelerate training, we omit gradient computation for the linear solves, including both the advection linear solve and the Poisson solver used in the PISO solver. The total training time is approximately 42 hours on a single NVIDIA RTX 2080 Ti GPU.

# I  Extended results

## I.1  CSM **results**

Conceptually, CSM shares the same spirit as SITL: it applies the learned correction directly to the coarse solver's output, but scales the correction by the time-step $\Delta t$. To provide a more direct

comparison, we have run additional experiments on both the Kuramoto–Sivashinsky (KS) and Backward-Facing Step (BFS) cases. We evaluate each method using the same metrics as in section 6, specifically $R^2$ for KS and MSE for the BFS. Since CSM only slightly differs from SITL, it also remains susceptible to perturbation growth, particularly in chaotic regimes where small errors can rapidly magnify over long-term rollouts. The quantitative results in table 2 confirm INC's superiority across all network architectures on the KS system and in the BFS flow. For the KS problem, INC achieves $R^2$ scores markedly higher than the corresponding CSM results, and a roughly 42% relative increase in $R^2$ has been observed with the DeepONet model. In the BFS case, the flow separation and reattachment zones are inherently unstable, where perturbations can trigger significant changes in such zones by affecting the main flow and the shear layer. INC attains an MSE of $(0.24 \pm 0.07) \times 10^{-3}$, compared to $(8.09 \pm 1.52) \times 10^{-3}$ for CSM, an improvement in error magnitude by a factor of approximately 33. These results reinforce the effectiveness of our indirect correction strategy.

Table 2: Comparison of $R^2$ (larger is better, used for KS metrics) and MSE (smaller is better, used for BFS) across models.

| | $R^2$ ($\uparrow$) | | | | MSE ($\downarrow$) |
|---|---|---|---|---|---|
| | KS(FNO) | KS(DeepONet) | KS(UNet) | KS(ResNet) | BFS(CNN) |
| INC | $0.97 \pm 0.08$ | $0.91 \pm 0.14$ | $0.97 \pm 0.06$ | $0.93 \pm 0.13$ | $(0.24 \pm 0.07) \times 10^{-3}$ |
| CSM | $0.90 \pm 0.17$ | $0.64 \pm 0.34$ | $0.89 \pm 0.19$ | $0.72 \pm 0.32$ | $(8.09 \pm 1.52) \times 10^{-3}$ |

## I.2   Modern-UNet results

We integrated a Modern-UNet [62] with our INC method and conducted comparative experiments. The results of these new experiments are fully consistent with the findings reported for other networks in the manuscript, as listed in table 3. Among the hybrid methods, INC achieves the lowest error rates, with a reduction of ca. 70% compared to SITL. Doubling the spatial resolution to 32 points further accentuates INC's advantage, representing an approximate 84% reduction relative to SITL. Similar to the results in the manuscript, the hybrid-solver approaches also clearly outperform the purely data-driven RNN method.

Table 3: Comparison of methods on Burgers of Morden-Unet (MSE).

| | MSE ($\downarrow$) | | | |
|---|---|---|---|---|
| | INC | SITL | SITL$^*$ | RNN |
| Burgers-Res16 ($\times 10^{-2}$) | $1.24 \pm 1.43$ | $4.62 \pm 3.31$ | $4.31 \pm 3.01$ | $334.98 \pm 79.78$ |
| Burgers-Res32 ($\times 10^{-2}$) | $0.29 \pm 0.46$ | $1.81 \pm 1.83$ | $1.97 \pm 1.99$ | $355.41 \pm 112.18$ |

## I.3   Physical metrics

For clarity, the evaluation of most cases is limited to standard statistical metrics in this manuscript, except for the TCF. In the following, we present a dedicated analysis tailored to each configuration. For the KS system, the energy spectrum is examined to identify the characteristic modes of the cascade. The energy is defined as

$$E_n = \tfrac{1}{2}|\hat{u}_n|^2, \tag{I.1}$$

where $\hat{u}_n(t)$ is the velocity field in Fourier space, which can be expressed as

$$\hat{u}_n(t) = \frac{1}{L} \int_0^L u(x,t)\, e^{-ikx}\, dx, \qquad k = \frac{2\pi n}{L}, \tag{I.2}$$

where $u_n(t)$ is the velocity and $L$ is the domain length.

For the Karman case, the viscous drag coefficient quantifies the tangential shear contribution to the total aerodynamic drag. The force acting on the obstacle surface $\Gamma$ is given by

$$F_d^{\text{viscous}} = \int_\Gamma \boldsymbol{\tau}_w \cdot \boldsymbol{t}_x\, d\Gamma, \tag{I.3}$$

where $\boldsymbol{\tau}_w = \mu \left( \nabla \boldsymbol{u} + \nabla \boldsymbol{u}^\mathsf{T} \right) \cdot \boldsymbol{n}$ is the viscous stress vector acting on the surface, $\mu$ is the dynamic viscosity, $\boldsymbol{n}$ denotes the local unit normal pointing into the fluid, and $\boldsymbol{t}_x$ is the unit vector aligned with the streamwise direction. The corresponding viscous drag coefficient is defined as

$$C_d^{\text{viscous}}(t) = \frac{1}{\frac{1}{2}\rho U_{\text{ref}}^2 \Gamma} \int_\Gamma \mu \left[ \left( \nabla \boldsymbol{u} + \nabla \boldsymbol{u}^\mathsf{T} \right) \cdot \boldsymbol{n} \right] \cdot \boldsymbol{t}_x \, d\Gamma, \tag{I.4}$$

where $\rho$ is the fluid density, $U_{\text{ref}}$ a reference velocity, and $\Gamma$ the total surface area.

For the BFS case, wall-shear dynamics govern the flow separation, recirculation, and reattachment processes. Accordingly, the local skin-friction coefficient is defined as

$$C_f(x, t) = \frac{\tau_w(x, t)}{\frac{1}{2}\rho U_b^2}, \tag{I.5}$$

where $\rho$ is the fluid density and $U_b$ denotes the bulk velocity. The wall shear stress $\tau_w$ is obtained from the tangential velocity gradient at the wall,

$$\tau_w(x, t) = \mu \left. \frac{\partial u_t}{\partial n} \right|_{\text{wall}}, \tag{I.6}$$

with $\mu$ the dynamic viscosity, $u_t$ the tangential velocity component, and $n$ the coordinate normal to the wall. This formulation enables direct evaluation of the instantaneous wall-shear distribution, providing quantitative insight into the near-wall momentum exchange and reattachment behavior downstream of the step.

The mean squared errors (MSE) of the statistical quantities are summarized in table 4.

Table 4: MSE of case-specific physical statistics for different methods.

|  | INC | SITL | SITL$^*$ |
| --- | --- | --- | --- |
| KS ($\times 10^{-4}$) | $1.49 \pm 5.94$ | $34.94 \pm 43.82$ | $23.76 \pm 28.70$ |
| BFS ($\times 10^{-3}$) | $6.05 \pm 0.57$ | $21.75 \pm 1.06$ | $23.34 \pm 2.91$ |
| Karman ($\times 10^{-2}$) | $1.87 \pm 1.54$ | $5.70 \pm 5.17$ | $8.48 \pm 6.09$ |

