# OpenReview forum: "INC: An Indirect Neural Corrector for Auto-Regressive Hybrid PDE Solvers"
_NeurIPS.cc/2025/Conference — NeurIPS 2025 poster_

### Official Review · Reviewer_MxgN · 2025-06-28

**Clarity:** 3
**Significance:** 2
**Originality:** 2
**Rating:** 5
**Confidence:** 4

**Summary:**

This paper Introduces Indirect Neural Corrector (INC), a neural corrector implemented as a source term in the partial differential equation. Authors provide a perturbation analysis justifying the superiority of the indirect method compared to direct correction term which directly adjusts the field. Experiments are conducted over a wide range of partial differential equations (PDEs) to provide empirical support of the proposed framework. These experiments support the theoretical analysis, and demonstrates up to ~158% improvement on the long trajectory performance.

**Questions:**

1) How is this paper different than other neural correctors implemented as source term? (Read W1) Addressing this can significantly help the readers to better understand the novelties.

2) Can authors discuss further if the function calls would be any different between the direct and indirect correction methods?

Please refer to the weakness for other questions

**Ethical Concerns:**

["NO or VERY MINOR ethics concerns only"]

**Final Justification:**

After considering the authors rebuttal and other reviewers comments I decided to keep my score. I believe this is a strong paper providing theoretical evidence of their methods superiority. Although the novelty of the method itself can be questioned further I believe the theoretical justification puts this research in a position where accepting would add value to the conference and the community.

**Limitations:**

yes

**Paper Formatting Concerns:**

No formatting concerns

**Quality:**

3

**Strengths And Weaknesses:**

**Strengths**
1. The necessity of hybrid neural solvers are well justified and argued in the paper
2. INC directly addresses the error accumulation issues of the hybrid neural solvers
3. The perturbation analysis is very well written and correctly reaches the point. This seems to be a novel analysis which is very well structured to demonstrate why adding the correction terms as a source term yields significant improvements.
4. A wide range of experiments are conducted across 6 different PDEs to support the claims. This provides a solid ground to observe generalization of the framework as well as the limitations.

**Weaknesses**
1. The major concern regarding this paper is the novelty of the proposed framework itself. The Indirect Neural Corrector performs the corrections inside the equation as a source term. This similar approach is used in other literature as well. Some of these approaches are not acknowledged by the authors. One example of this is "Deep neural network enabled corrective source term approach to hybrid analysis and modeling" [1].

2. The theoretical analysis is only conducted for the backward Euler case, meanwhile in the experimental configuration authors discuss the difference in implementation of INC for different time stepping schemes. Can authors discuss this further and explain what was the difference between implementation INC for backward Euler, and other schemes like Runge-Kutta?

3. The learning dynamics is not described well. The authors need to explain how the model is trained in more detail. For instance, they mention the training is done in a supervised manner, does this mean the model was trained separately and then used online with the coarse solver?

4. The effect of multi-step unrolling strategy is something that is completely neglected. Does the framework backpropagate through all the temporal steps or just a few steps? This can lead to significant inefficiencies that authors must acknowledge or briefly discuss.

5. For clarity purposes authors must clarify the notations better at the beginning of the theoretical justification.


(W1 Remains the primary weakness for the reviewer)

**Ref**

[1] Blakseth, S.S., Rasheed, A., Kvamsdal, T. and San, O., 2022. Deep neural network enabled corrective source term approach to hybrid analysis and modeling. Neural Networks, 146, pp.181-199.

---

> ### Author Rebuttal · Authors · 2025-07-30
>
> We sincerely thank the reviewer for their thoughtful and constructive review. We appreciate the acknowledgment of our work's theoretical contributions in tackling long-term error accumulation in hybrid neural solvers and for highlighting the strengths of our analysis and the  broad experimental validation. Meanwhile, we welcome the reviewer's concerns, particularly regarding the novelty in relation to existing correction approaches, clarity on learning dynamics, and the handling of unrolling strategies. In what follows, we address each of the raised weaknesses and questions in detail.
>
> ## Questions and Weaknesses
>
> > W1 / Q1: How is this paper different than other neural correctors implemented as source term? Addressing this can significantly help the readers to better understand the novelties.
>
> We thank the reviewer for highlighting this limitation and appreciate the reference to the CoSTA work[1]. We will discuss similarities and differences in the camera-ready version. CoSTA, similar to INC, uses a neural network to learn a corrective source term for a physics-based model. However, there are several differences in scope and direction compared to our submission:
>
> (1) **Long‑Term Stability**: Whereas CoSTA trains a DNN for single‑step corrections on a 1D diffusion problem, INC is explicitly formulated and trained for autoregressive multi-step rollouts and evaluated for long time periods.
>
> (2) **Analytical contribution:** We derive a closed-form error-propagation theoretical analysis, which directly addresses and resolves the issue of error accumulation in multi‑step simulations.
>
> (3) **Breadth and generality:** INC is demonstrated on 6 different cases from 1D to 3D, using four distinct network families and three differentiable solvers. To the best of our knowledge, no previous study shows comparable scope or architecture-agnostic performance.
>
> > W2: The theoretical analysis is only conducted for the backward Euler case, meanwhile in the experimental configuration authors discuss the difference in implementation of INC for different time stepping schemes. Can authors discuss this further and explain what was the difference between implementation INC for backward Euler, and other schemes like Runge-Kutta?
>
> > Q2: Can authors discuss further if the function calls would be any different between the direct and indirect correction methods?
>
> We thank the reviewer for raising this important point about the implementation details. Our theoretical analysis is indeed conducted in the setting of the forward Euler scheme, as it allows for a clean derivation and explicit characterization of the error amplification behavior. However, our INC is not limited to this scheme in practice. In our experiments, we demonstrate that INC can be adapted to more sophisticated time-integration methods, including Runge-Kutta and a backward Euler scheme in PISO. By extending the INC framework to broader numerical schemes, we studied INC’s broader adaptability to diverse configurations. Regarding function calls and modular implementation, the detailed algorithmic implementations with accompanying mathematical formulations are provided in Appendices E, F, and G for Forward Euler, Runge-Kutta, and the Backward Euler of PISO, respectively. Below, we summarize the core ideas, and we will clarify them in the camera-ready version of our paper.
>
> (1) **Forward Euler Scheme:** The INC correction term $G_{\theta}(u)$ is incorporated as an additional term on the RHS of the governing PDE. The numerical solver thus updates the solution using the standard Euler integration step, inherently combining both the physical dynamics and the learned neural correction term within a straightforward integration step.
>
> (2) **Exponential Time Differencing Runge-Kutta (ETDRK) Scheme:** In the more advanced ETDRK scheme combined with the spectral method, INC correction $G_{\theta}(u)$ is handled differently but still intricately integrated into the numerical solver's logic itself. For the ETDRK implementation, the learned correction $G_{\theta}(u)$ is explicitly fused with the nonlinear term within Fourier space prior to applying the ETDRK time-stepping integration.
>
> (3) **Backward Euler Scheme for Pressure-Implicit with Splitting of Operators (PISO):** In the PISO with Backward Euler scheme, the INC correction term is embedded into the momentum equation as a right-hand side term. This integration means that the correction continuously influences both the advection step and the iterative pressure correction process of the PISO algorithm. This "deeper" incorporation ensures that the INC method effectively stabilizes and guides the solver through complex pressure-velocity coupling.
>
> Regarding the function calls:
>
> (1) **Direct Correction (SITL/SITL\*):** The neural network correction is simpler in terms of function call structure, as it is directly added to the solution either immediately after (SITL) or just before (SITL*) each numerical solver step. Thus, each call to the neural network stands separate from the numerical integration, rather than being embedded into it.
>
> (2) **Indirect Correction (INC Framework):** In contrast, the INC framework inherently integrates the neural network call within each numerical solver iteration. Instead of being an external, post-processed correction, the INC correction is inherently embedded within the numerical scheme's iterative loop or temporal discretization. This embedding within the solver's iteration modifies the structure and timing of neural network function calls, linking neural evaluations to solver iterations. According to our study, this structural integration is central to the enhanced stability and accuracy.
>
>
> >W3: The learning dynamics is not described well. The authors need to explain how the model is trained in more detail. For instance, they mention the training is done in a supervised manner, does this mean the model was trained separately and then used online with the coarse solver?
>
> We thank the reviewer for highlighting the need for a clearer explanation of the learning dynamics and training process of our model. We will ensure this important distinction is more explicitly emphasized in the revised manuscript.
>
> To clarify, the INC model is trained with the solver **simultaneously in a supervised manner using a multi-step unrolled optimization strategy**. Specifically, the training data is generated from high-fidelity simulations and downsampled to the low resolution. During the training phase, the neural corrector network is optimized to minimize the discrepancy between trajectories produced by the hybrid solver (coarse numerical solver combined with INC) and these downsampled high-fidelity solutions. The loss function is typically an L2 metric computed over multi-step autoregressive rollouts. This multi-step training is conducted in an end-to-end differentiable framework (all solvers are implemented in PyTorch and CUDA), allowing gradients to backpropagate through both the neural corrector and the numerical solver steps for the whole rollout. This training paradigm ensures that the neural corrector learns corrections optimized explicitly for long-term stability and accuracy, and accounts for the solver's intrinsic numerical dynamics. The number of steps unrolled at training time varies and is detailed in Appendix H. In the inference phase, the trained model is coupled with the low-resolution solver simultaneously to run the simulation.
>
> >W4: The effect of multi-step unrolling strategy is something that is completely neglected. Does the framework backpropagate through all the temporal steps or just a few steps? This can lead to significant inefficiencies that authors must acknowledge or briefly discuss.
>
> Thank you for pointing out the lack of clarity here: Unrolling over multiple steps in training improves inference performance, but increases the memory requirements of saving neural network activations inside each unrolled time step in the forward pass. These trade-offs and benefits of unrolling strategies have been studied by other researchers [2,3]. In our work, the choice of unrolling length is guided by the characteristic timescale $t_{c}$ of the dynamical systems.
>
> For each case, $t_{c}$ is derived from system-specific properties, like the inverse of the maximum Lyapunov exponent for the Kuramoto–Sivashinsky equation. More details about $t_{c}$ for other cases are listed in Appendix H. After the calculation of $t_c$ and setting the time step as $\Delta t$, we compute the number of steps per characteristic timescale as $N=\frac{t_c}{\Delta t}$. We then set the unrolling length during training to approximately $0.04\cdot N$.  During inference, we significantly extend this horizon to about 100 times longer, typically 4$\cdot N$ ~ 6$\cdot N$, to rigorously evaluate long-term stability and accuracy. We believe that anchoring both training and inference lengths to the system's intrinsic timescale results in a more physically grounded and robust learning process.
>
>
> >W5: For clarity purposes authors must clarify the notations better at the beginning of the theoretical justification.
>
> We thank the reviewer for this suggestion. In the revised manuscript, we will reorganize the section on theoretical justifications to improve readability and introduce symbols and operators in a “Notation” table.
>
> ---
> References:
>
> [1] Blakseth S S, et al. Deep neural network enabled corrective source term approach to hybrid analysis and modeling[J]. Neural Networks, 2022
>
> [2] Kochkov, D. et al. Machine learning–accelerated computational fluid dynamics[J]. Proceedings of the National Academy of Sciences, 2021
>
> [3] List, B. et al. Differentiability in unrolled training of neural physics simulators on transient dynamics[J]. Computer Methods in Applied Mechanics and Engineering, 2025

---

### Official Review · Reviewer_Gzsb · 2025-07-01

**Clarity:** 3
**Significance:** 2
**Originality:** 2
**Rating:** 2
**Confidence:** 4

**Summary:**

This paper introduce An Indirect Neural Corrector for Auto-Regressive Hybrid PDE Solvers,which integrates learned corrections into the governing equations rather than applying direct state updates.They claim that INC reduces the error amplification on the order of  $\Delta t^{-1} + L$. And poses no architectural requirements and integrates seamlessly with arbitrary neural networks and solvers.They evaluate INC on 1D to 3D cases, and achieve the improvement compare with baselines.

**Questions:**

1. Please clarify the conceptual and methodological differences between the proposed Indirect Neural Corrector and prior approaches such as P2C2Net , which also utilize neural networks to correct coarse-grid errors over time.
2. Would applying the proposed method to more advanced backbone models—such as Learned Correction (LC) [1]  or PDE-Refiner—lead to further performance improvements, particularly on challenging benchmarks like Kolmogorov flow?
3. The current evaluation is limited to standard statistical metrics. It would be beneficial to include physically meaningful metrics, such as the energy spectrum curve in turbulent flows, to provide a more comprehensive assessment.
4. While hybrid solvers offer interpretability and some physical grounding, they often ignore historical information and tend to be slower than neural operator-based methods. Incorporating operator-based techniques could potentially lead to significant speed-ups and broader applicability. Such integration would represent a promising and impactful research direction.

***Ref***

[1] Kochkov et al. Machine learning–accelerated computational fluid dynamics. PNAS, 2021.

**Ethical Concerns:**

["NO or VERY MINOR ethics concerns only"]

**Final Justification:**

I apologize for the delayed response; I mistakenly believed that clicking 'Mandatory Acknowledgement' concluded the conversation, and did not receive a follow-up email.

While I appreciate the authors' rebuttal, I am not convinced by their arguments.

The central issue remains the conceptual novelty of the correction mechanism. To my knowledge, whether a correction is applied indirectly or directly to the right-hand side (RHS) of the equations, the underlying principle is the same. The distinction of whether the network's input features are combined before or after this correction is, in my view, an architectural detail rather than a substantive innovation.

Consequently, the novelty of this core idea seems insufficient for a top-tier conference like NeurIPS. This approach bears similarities to work such as Xu et al. [1], and raises concerns that the contribution is more opportunistic in its design than fundamentally novel.

Given these points, I will be lowering my score.

**Ref**

[1] Xu et al. FP64 is All You Need: Rethinking Failure Modes inPhysics-Informed Neural Networks, 2025

**Limitations:**

- The proposed method is heavily dependent on the underlying numerical solver, which may limit its generalizability across different simulation settings or solver configurations.
- The approach appears tailored specifically for hybrid solver frameworks and may not be easily applicable to purely data-driven or operator-based methods.

**Paper Formatting Concerns:**

- Figures are referenced inconsistently (e.g., fig. 2, fig. 4a, fig. 3)
- Missing Equation Parentheses (e.g., Eq. (9))

**Quality:**

3

**Strengths And Weaknesses:**

### Strengths
- The proposed method demonstrates improved performance over established baselines by leveraging the idea of an Indirect Neural Corrector (INC).
- The model is conceptually simple and demonstrates potential in correcting low-resolution numerical solutions, which is practically relevant given the high computational cost of traditional solvers.

### Weaknesses
- The idea of an indirect neural corrector may not be entirely novel. For instance, P2C2Net employs a similar strategy—computing derivatives on coarse grids and correcting errors over time using neural networks, with strong generalization results on Kolmogorov flow.
- Some relevant baselines are missing from the comparisons, such as PDE-Refiner [2] and Modern-UNet [3], which are important references for learning-based PDE solvers.
- Although the authors claim model-agnosticism, including one or two concrete architectural examples would greatly enhance clarity and reader comprehension.
- Details regarding the numerical data generation process are lacking, including time discretization schemes, step sizes, and grid parameters, which are crucial for reproducibility and fair evaluation.

***Ref***

[1] Wang et al. P2C2Net: PDE-Preserved Coarse Correction Network for Efficient Prediction of Spatiotemporal Dynamics. NeurIPS, 2024.

[2] Lippe et al. PDE-Refiner: Achieving Accurate Long Rollouts with Neural PDE Solvers. NeurIPS, 2023.

[3] Gupta et al. Towards Multi-spatiotemporal-scale Generalized PDE Modeling. TMLR, 2023.

---

> ### Author Rebuttal · Authors · 2025-07-30
>
> We thank the reviewer for the thoughtful feedback on our manuscript. We are especially grateful for the recognition of the interpretability and physical grounding of our method, and the performance improvements over established baselines. We carefully considered the reviewer's concerns, and provide a detailed response, along with additional experiments and metric evaluations that further support the claims and generality of our approach. Regarding the formatting issues, we will revise them in the camera-ready version.
>
> ## Questions and Weaknesses
>
> > W1: The idea of an indirect neural corrector may not be entirely novel. For instance, P2CNet employs a similar strategy---computing derivatives on coarse grids and correcting errors over time using neural networks, with strong generalization results on Kolmogorov flow.
> > Q1: Please clarify the conceptual and methodological differences between the proposed Indirect Neural Corrector and prior approaches such as P2CNet, which also utilize neural networks to correct coarse-grid errors over time.
>
>
> P2C2Net represents a physics-encoded model that uses a learnable filter to estimate spatial derivatives, integrates them with an RK4 scheme, and then applies a direct neural correction to the solution. This is similar to papers like [1]. In contrast, our INC framework introduces its correction as a learnable RHS term within the governing PDE. This indirect correction is coupled within a full numerical scheme for both temporal and spatial derivatives, which provably dampens the amplification of perturbations over long rollouts. The numerical solver at its core ensures that the trajectory is constrained physically. Nonetheless, extending the derivation, and integrating an INC-like scheme into a method like P2CNet would be a very interesting direction for future work.
>
> > Q2: Would applying the proposed method to more advanced backbone models---such as Learned Correction (LC) [1] or PDE-Refiner---lead to further performance improvements?
>
> We thank the reviewer for raising the applicability of our method to stronger baselines. Below, we address this point in more detail.
>
> The comparison between INC and LC is actually **provided by the comparison of INC and SITL in our paper**. SITL and LC are the same, where a coarse numerical solver $\mathcal{T}$ firstly advances the state, producing an intermediate solution $u_n^{\*}=\mathcal{T}(u_n)$. Then a neural network $G_\theta$ directly corrects that solver output to yield the next state: $u_{n+1} = G_\theta(u_n^{\*})$, where a residual connection has been applied such that $G_\theta(u_n^{\*})=u_n^{\*}+\theta(u_n^{\*})$.  Most importantly, our results show that methods like LC/SITL are inferior to INC, e.g., INC reduces the MSE by up to 97.1\% in the challenging backward-facing step case.
>
> *PDE-Refiner* is more closely related to generative models with a **stochastic process**: they predict the full state of the system in each time step. The context of INC (also LC and SITL above) are **deterministic processes** where the coarse-grid simulation is "nudged" back towards the correct trajectory at each time step. The combination of these corrections with stochastic learning approaches would be an interesting direction for future work, but it's unfortunately beyond our current scope.
>
> > W2: Some relevant baselines are missing from the comparisons, such as PDE-Refiner [2] and Modern-UNet [3], which are important references for learning-based PDE solvers.
> > W3: Although the authors claim model-agnosticism, including one or two concrete architectural examples would greatly enhance clarity and reader comprehension.
>
> Thank you for suggesting a more recent network architecture like the Modern-UNet. We followed your suggestion and **integrated a Modern-UNet with our INC method** and conducted comparative experiments against several baselines: an LC (SITL) model, the pre-correction variant (SITL*), and an RNN. The results of these new experiments are fully consistent with the findings reported for other networks in the paper. Among the hybrid methods, **INC achieves the lowest error rates, a reduction of ca. 70\% compared to LC/SITL**. Doubling the spatial resolution to 32 points further accentuates INC’s advantage, **representing an approximate 84\% reduction relative to LC/SITL**. Similar to the results in our paper, the hybrid‐solver approaches also very clearly outperform the purely data‐driven RNN method.
>
>
> ||INC|SITL|SITL*| RNN |
> |-|-|-|-|-|
> |Burgers-Res16 ($\times10^{-2}$)|$1.24 \pm 1.43$|$4.62 \pm 3.31$|$4.31 \pm 3.01$|$334.98 \pm 79.78$|
> |Burgers-Res32 ($\times10^{-2}$)|$0.29 \pm 0.46$|$1.81 \pm 1.83$|$1.97 \pm 1.99$|$355.41 \pm 112.18$|
>
> > Q2: ... performance improvements, particularly on challenging benchmarks like Kolmogorov flow?
>
> Actually, we intentionally excluded the Kolmogorov flow scenario to **focus on more challenging turbulence cases: the 2D Karman, 2D Backward-Facing Step, and 3D Turbulent Channel Flow**.
> 1) those cases **cover all important boundary conditions** including no-slip, periodic, Neumann, and Dirichlet.
> 2) external forcing by mean flow or pressure gradients produces inhomogeneous fields. The interaction between walls and inhomogeneity generates turbulence through boundary layer separation, shear, or friction, also makes the system **more prone to perturbations**.
> 3) all the cases are **resolution-sensitive**, requiring fine grids near walls to capture boundary layers, which is the key to wall-bounded turbulence. When we employ a coarse grid, it amplifies unstable shear layers, significantly changing the structure of the field [2]. They are standard in stability studies [3].
>
> In contrast, the Kolmogorov flow, a homogeneous turbulent flow in periodic domains without solid boundaries, is less sensitive to localized perturbations, with its statistical homogeneity limiting spatial amplification and primarily affecting averages rather than local dynamics. Thus, our selection highlights INC’s potential for real-world scenarios.
>
> > Q3: The current evaluation is limited to standard statistical metrics. It would be beneficial to include physically meaningful metrics, such as the energy spectrum curve in turbulent flows, to provide a more comprehensive assessment.
>
> We thank the reviewer for the suggestion to include physically meaningful metrics. In response, we have incorporated a dedicated analysis tailored to each configuration. For the KS system, we examine the energy spectrum to capture the characteristic modes of the cascade. In the Karman case, we assess the aerodynamic performance via the viscous drag coefficient, calculated as $C_d^{\text{viscous}} = \frac{F_d^{\text{viscous}}}{\frac{1}{2} \rho U_{\text{ref}}^2 \Gamma}$, where $ F_d^{\text{viscous}} = \int_{\Gamma} \mathbf{\tau} \cdot  \hat{\mathbf{n}} \cdot \hat{\mathbf{x}}~d\Gamma$. In the BFS case, wall shear dynamics govern separation, recirculation, and reattachment. So, we evaluate the wall skin‐friction coefficient, calculated as $C_f = \frac{\tau_w}{\frac{1}{2} \rho U_{b}^2}$, where $ \tau_w = \mu \left. \frac{\partial{u}}{\partial{y}} \right|_\text{wall}$. Herein, we present the MSE of statistics detailed in tables.
>
> ||INC|SITL|SITL*|
> |-|-|-|-|
> |KS ($\times 10^{-4}$)|$1.49 \pm 5.94 $|$34.94 \pm 43.82$|$23.76 \pm 28.70$|
> |BFS ($\times 10^{-3}$)|$6.05 \pm 0.57$|$21.75 \pm 1.06$|$23.34 \pm 2.91$|
> |Karman ($\times 10^{-2}$)|$1.87 \pm 1.54$|$5.70 \pm 5.17$|$8.48 \pm 6.09$|
>
> Across all the cases, **INC reduces the MSE by a substantial margin compared to the direct correction methods**. In the KS case, INC’s MSE is more than an order of magnitude lower than LC/SITL, indicating substantially improved spectral fidelity. For BFS, the skin‐friction MSE is reduced by ca. 72\% with INC compared to LC/SITL. This demonstrates a significantly more accurate reproduction of the wall‐shear dynamics. Finally, in the Karman case, INC’s MSE outperforms LC/SITL by 67\%, showing a superior reconstruction of viscous drag characteristics. The results confirm that **INC substantially outperforms both direct‐correction baselines**, aligning more closely with the physically meaningful quantities that govern complex flow behavior.
>
> > Q4: While hybrid solvers offer interpretability and some physical grounding, they often ignore historical information and tend to be slower than neural operator-based methods. Incorporating operator-based techniques could potentially lead to significant speed-ups and broader applicability. Such integration would represent a promising and impactful research direction.
>
> Thank you for highlighting this important trade-off. While hybrid solvers incur higher runtimes compared to neural operator-based surrogates, they offer interpretability and the output is constrained by physical laws, which makes them very attractive to practitioners. Notably, although it's less efficient than purely neural operator-based surrogates, **when targeting equivalent levels of accuracy**, traditional numerical solvers commonly used in industry incur **computational costs up to 300 times higher than those of INC**.
>
>
> >W4:Details regarding the numerical data generation process are lacking, including time discretization schemes, step sizes, which are crucial for reproducibility and fair evaluation.
>
> Thank you for pointing this out. While the numerical setup is in Appendix H and discretization schemes are in Section 5, we agree that spreading details across sections has reduced clarity. In the revision, we will consolidate key data generation details into a dedicated section to improve reproducibility and accessibility.
>
>
> ---
> References:
>
> [1] Rao C, et al. Encoding physics to learn reaction–diffusion processes[J]. Nature Machine Intelligence, 2023
>
> [2] Schmid P J, et al. Stability and transition in shear flows[M]. Springer Science & Business Media, 2012
>
> [3] Schäfer F, et al. The dynamics of the transitional flow over a backward-facing step[J]. Journal of Fluid Mechanics, 2009

---

> > ### Comment · Reviewer_Gzsb · 2025-08-04
> > **Justifying INC's Novelty Beyond P2C2Net**
> >
> > I appreciate the authors' efforts during the rebuttal period. However, I believe the response did not adequately address my core concern regarding novelty. To the best of my knowledge, both methods share essential similarities:
> > 1. **Physics-computation core**:
> >    - P2C2Net’s physics block is a learnable, differentiable framework that computes the RHS.
> >    - INC’s coarse solver computes intermediate or coarse solutions, which are conceptually analogous to RHS computation.
> > 2. **Per-step correction**:
> >    - P2C2Net’s neural network block uses feature terms (derivatives, velocity/pressure fields) to correct coarse solutions, *which inherently define the RHS*.
> >    - INC’s network $\mathcal{G}_\theta$ corrects intermediate states, *but the state updates implicitly modify the RHS*.
> >
> > **Fundamental issue**: The core idea—embedding a neural corrector into a physics solver—appears functionally equivalent. The differences (state vs. feature correction) seem superficial, as both ultimately adjust the RHS *via neural networks at every rollout step*.
> >
> > You need to justify your claimed novelty, given that the core idea is essentially the same as that of P2C2Net, and demonstrate that these seemingly minor differences lead to measurable improvements over P2C2Net.

---

> > > ### Author Response · Authors · 2025-08-05
> > >
> > > We appreciate the reviewer's feedback. While both INC and P2C2Net aim to integrate neural networks into "solvers" as other hybrid neural solvers, their motivations and mechanisms are substantially different.
> > >
> > > The core objective of INC is to **improve numerical stability when coupling neural networks with numerical solvers**, which requires a strict adherence to numerical stability and is particularly sensitive to perturbations. This limitation has also been acknowledged in the P2C2Net paper, where it is circumvented by **replacing** the strict numerical scheme with a learnable kernel. It's an interesting idea. However, as the reviewer rightly noted, "the numerical scheme provides a solid physical grounding and clear interpretability", which becomes increasingly essential in complex systems, such as long-term 3D turbulence, **where P2C2Net has not yet been applied**.
> > >
> > > Although INC and P2C2Net share the same goal as other hybrid neural solvers, the key differences are:
> > >
> > > ### 1. The Core Mechanism: Indirect correction within the numerical solver (INC) vs. Direct correction with learnable kernel (P2C2Net)
> > >
> > > For the reviewer's core concern, "state vs. feature correction seem superficial", is factually not true, especially when the correction is integrated with numerical solvers.
> > >
> > > *   **P2C2Net employs a *Direct State Correction* with learnable kernel.** As the P2C2Net paper states (Sec. 3.2.3), the model first computes a "neural-corrected coarse solution state $\hat{u_k} = \text{NN}(u_k)$". This corrected state is then fed into their learnable kernel to calculate derivatives. **This is a direct correction of the state variable before it enters the learnable kernel.**  **This is functionally identical to the SITL\* baseline in our paper, which we explicitly benchmark against and show stability limitations.** While P2C2Net uses a learned kernel in place of a numerical method, it bypasses the numerical stability requirement at the cost of losing the physical grounding and interpretability. For this reason, performance comparisons with SITL* cannot be directly extended to P2C2Net.
> > >
> > > *   **INC implements an *Indirect RHS Correction* within full numerical scheme.** In contrast, since the numerical scheme is prone to perturbations, INC does not directly alter the state $u_n$, but influences $u_n$ indirectly through the solver’s dynamics. The full numerical scheme firstly applied to compute the derivatives. Network then provides an *additive correction term* on the RHS of PDEs, which affects the solution indirectly by going through the dynamics. This architectural difference is the direct cause of INC's superior stability compared with other hybrid methods, which couple with a numerical solver.
> > >
> > > ### 2. Broader Applicability and Validation Across Complex Systems
> > >
> > > The robust theoretical foundation of INC and the kept numerical scheme translate directly to measurable improvements and broader applicability.
> > >
> > > * **Limitations of P2C2Net:** As the "solver" employed in P2C2Net is a learnable kernel, rather than a general-purpose numerical scheme with physical grounding. This design imposes notable limitations. As acknowledged by the original paper, the model is based on regular grids with periodic boundaries, and only validated for 2D systems.
> > >
> > > *   **Generality of INC:** INC’s framework is both solver-agnostic and model-agnostic. This allows it to couple with diverse, off-the-shelf differentiable solvers and networks. Our experiments demonstrate INC's success across **1D, 2D, and 3D systems, handling diverse boundary conditions (periodic, Dirichlet, Neumann, and no-slip) and grid structures (regular and refined)**. Meanwhile, the enhanced stability allows INC to operate with much larger time steps (i.e., higher CFL numbers) where direct methods like SITL* would fail or suffer severe accuracy degradation. This is precisely why INC achieves significant speed-ups (e.g., **330x for 3D turbulence**) while maintaining accuracy, it can take larger steps without becoming unstable.
> > >
> > > The novelty of INC is not the general hybrid neural solver concept, but the **specific, theoretically-grounded framework of integrating neural correction with numerical solvers in a stable way**. Unlike prior methods, which directly couple neural networks to solver outputs or replace the solver altogether (as in P2C2Net), INC offers a principled, stable, and generalizable alternative. We definitely appreciate the contributions of P2C2Net, which follows a new track by replacing the numerical scheme with a learnable kernel. The theoretical proof in our paper holds for classic numerical solvers, where both spatial and temporal discretization are well-defined to ensure the physical law is respected explicitly. Thus, whether our theoretical proof covers learned kernels like P2C2Net is an open question for future research. Nonetheless, our paper makes important contributions to the wide range of learned correction methods that couple NNs with numerical solvers.

---

### Official Review · Reviewer_8Ecs · 2025-07-03

**Clarity:** 3
**Significance:** 2
**Originality:** 3
**Rating:** 4
**Confidence:** 3

**Summary:**

This paper aims to improve the long rollout accuracy of the hybrid solvers by adjusting the correction term from the original (after the numerical simulation) to the right-hand-side (RHS) part. This paper also provides the analytical analysis as well as numerical analysis about how accurate the proposed approach is compared to the conventional direct correction, where they defined the error dominance ratio. The proposed approach has been validated on different cases, from a 1D chaotic system to 3D turbulence.

**Questions:**

- In the benchmark comparison of Section 6.1, you compared SITL and SITL*. I understand that the previous one is the direct correction approach. But is not clear for readers about the latter one. Also, what are the specific differences between these two direct correction methods? Why do you need to include both of them as benchmarks?
- How do you choose the unrolling steps in the training and how it influence the final performance?
- How the INC method perform when applied to PDEs with stiff source terms? If you embed a neural network as part of the source term in a PDE, and the output of this network is potentially unbounded (e.g., not constrained by activation functions like sigmoid or tanh), could such an “uncontrolled” neural source introduce additional challenges to the stability of time integrators (such as Runge-Kutta or implicit solvers)?

**Ethical Concerns:**

["NO or VERY MINOR ethics concerns only"]

**Final Justification:**

Overall, the work is technically solid and has clear strengths in its theoretical framing and breadth of experiments. I keep my scoreing

**Limitations:**

- The author mentioned that for problems that are inherently strongly damped or use unconditionally stable formats, the advantages of INC may be weakened. This point is worth emphasizing, and users need to judge the applicability of INC according to the dynamic characteristics of their own problems and the numerical formats they use.
- The deployment of long-term sequence training requires a lot of computational resources, which may become a bottleneck in applying this method to ultra-large-scale problems.
- A large number of existing, mature scientific computing code libraries cannot be directly applied to the training paradigm of INC.

**Quality:**

3

**Strengths And Weaknesses:**

- Strengths:
    - Although the change of the proposed method to the direct correction approach is incremental, they provide a clear theoritical proof about why it works and how accurate it will be.
    - The paper provides sufficient examples to demonstrate the generality of the proposed method to different numerical solvers, neural networks, as well as different PDEs.
    - Comprehensive and convincing experiments support the claim and demonstrate the improvement of the proposed approach in terms of improving the long rollout accuracy.
    - The written and presentation are good.
- Weaknesses:
    - The paper mentioned the multi-step unrolled optimization to ensure the stability, which will cause a huge training cost, especially for 3D problems and long unrolling steps. However, it is not clear how to choose the best multi-step strategy. Another follow-up question is how the proposed method performs without this strategy?
    - The theoretical analysis (Section 4) is based on the linearized error evolution and the "small perturbation" assumption. In strongly chaotic or strongly nonlinear systems, the learned correction terms (i.e., perturbations) may not always be sufficiently small, and the system may deviate from the neighborhood where the linearized analysis is valid within a few steps. A further explanation on this would strengthen the paper.
    - The hybrid approach reply on the differential solvers. However, for many industry applications, the legacy solvers might not be differentiable, which will limit the implementation of the proposed approach.

---

> ### Author Rebuttal · Authors · 2025-07-30
>
> We thank the reviewer for the thoughtful feedback and for recognizing our theoretical contribution, generality across PDEs, and clear presentation. We appreciate the concerns regarding multi-step unrolling, the scope of our theoretical analysis, and reliance on differentiable solvers. Below, we clarify these points, including theoretical assumptions, benchmark variants (SITL vs. SITL*), the importance and choice of unrolling steps, and the way to bound the output of the neural network.
>
> ## Questions
>
> >Q1: In the benchmark comparison of Section 6.1, you compared SITL and SITL*. I understand that the previous one is the direct correction approach. But it is not clear for readers about the latter one. Also, what are the specific differences between these two direct correction methods? Why do you need to include both of them as benchmarks?
>
> Let us directly address the details on our baselines (SITL, SITL*) as follows.
>
> 1. Both SITL and SITL* are direct correction methods. The only difference is the order in which the direct correction and the solver operate. For SITL, the correction is applied after the solver[1,2]. A coarse numerical solver $\mathcal{T}$ firstly advances the state, producing an intermediate solution $\displaystyle u_{n}^{\*}=\mathcal{T}(u_n)$. Then a neural network $G_\theta$ **directly corrects** that solver output to yield the next state: $u_{n+1} = G_\theta\bigl(u_{n}^{\*}\bigr)$, where a residual connection has been applied that $G_\theta\bigl(u_{n}^{\*}\bigr)=u_{n}^{\*}+\theta(u_{n}^{\*})$. SITL* is a simple pre‑correction variant of the SITL. Instead of adding the learned correction after the solver update, SITL* applies the neural term before the solver step, thus $u_{n+1}=\mathcal{T}(u_{n}^{\*}),\quad u_n^\{*}=G_\theta\bigl(u_n\bigr)=u_n+\theta(u_n)$.
>
> 2. By evaluating both SITL and SITL*, we ensure our work covers all variants of direct methods. Thus, the results demonstrate that regardless of correction ordering, any **direct‑update method suffers from the same underlying error amplification $R_k$**.
>
> > W1:The paper mentioned the multi-step unrolled optimization to ensure the stability, which will cause a huge training cost, especially for 3D problems and long unrolling steps. However, it is not clear how to choose the best multi-step strategy. Another follow-up question is how the proposed method performs without this strategy?
> > Q2: How do you choose the unrolling steps in the training and how it influences the final performance?
>
> We thank the reviewer for raising this important question regarding the choice of unrolling steps during training and their influence on final performance.
>
> 1. Unrolling over multiple steps in training improves inference performance significantly, but increases the memory requirements since it is necessary to store neural network activations inside each unrolled time step in the forward pass [1,2,3].  In our work, the choice of **unrolling length is guided by the characteristic timescale $t_{c}$ of the dynamical systems**. For each case, $t_{c}$ is derived from system-specific properties, like the inverse of the maximum Lyapunov exponent for the Kuramoto–Sivashinsky equation. More details about $t_{c}$ for other cases are listed in Appendix H. After the calculation of $t_c$ and setting the time step as $\Delta t$, we compute the number of steps per characteristic timescale as $N=\frac{t_c}{\Delta t}$. We then set the unrolling length during training to approximately $0.04\cdot N$.  During inference, **we significantly extend this horizon to about 100 times longer**, typically 4$\cdot N$ to 6$\cdot N$, to rigorously evaluate long-term stability and accuracy. We believe that anchoring both training and inference lengths to the system's intrinsic timescale results in a more physically grounded and robust learning process.
>
> 2. Regarding the broader relationship between unrolling steps and performance, previous comprehensive analyses, such as those presented in List et al. [3], have systematically explored this issue. Given their detailed study, we opted not to replicate this investigation in our current work.
>
> > Q3: How the INC method performs when applied to PDEs with stiff source terms? If you embed a neural network as part of the source term in a PDE, and the output of this network is potentially unbounded (e.g., not constrained by activation functions like sigmoid or tanh), could such an "uncontrolled" neural source introduce additional challenges to the stability of time integrators (such as Runge-Kutta or implicit solvers)?
>
> We thank the reviewer for this insightful question regarding the potential stiffness introduced by unbounded neural network outputs when incorporated into PDEs. This issue is critical for the stability of hybrid solvers, not only for INC, but also for the direct correction method[1,2].
>
> 1. In our current setup, the neural network is treated as a regression model, and therefore, **no explicit output bounding** (e.g., via sigmoid or tanh activations) is imposed at the final layer. Activation functions are used only between internal layers, following standard practice for each backbone architecture: GeLU for FNO, tanh for DeepONet, and ReLU for CNN-based models. As a result, the output of the correction network is, in principle, unbounded.
>
> 2. In practice, we have observed symptoms of such stiffness when training hybrid solvers, particularly for more complex systems, such as the cases with our semi-implicit solver PISO. For instance, when training on the Karman case with a rollout length of 8 steps, training from scratch often leads to early divergence. This is due to the stiffness introduced when the poorly trained network’s output is injected into the solver, regardless of whether it is done directly or indirectly.
>
> 3. To mitigate this issue, we employ **a staged multi-step rollout strategy** as described in Appendix H. The model is first trained with a rollout length of 2 steps, followed by 8 steps. This progressive unrolling allows the network to gradually learn stable correction dynamics and reduces the risk of instability due to stiff or erratic outputs. Figure 4(a) in the paper further illustrates the importance of the learned correction in such settings. In the "No Model" baseline, which includes no learned correction, we observe clear oscillations caused by the low resolution and sharp corners. These oscillations eventually destabilize the simulation. In contrast, with INC correction, the system remains forward-stable throughout long rollouts, despite the lack of an explicit output constraint. Our approach does not enforce hard bounds on the neural network output. Instead, **by unrolling over multiple steps, the network learns to produce corrections in a physically consistent and numerically stable range**.
>
> >W2: The theoretical analysis (Section 4) is based on the linearized error evolution and the "small perturbation" assumption. In strongly chaotic or strongly nonlinear systems, the learned correction terms (i.e., perturbations) may not always be sufficiently small, and the system may deviate from the neighborhood where the linearized analysis is valid within a few steps. A further explanation on this would strengthen the paper.
>
> Regarding the assumptions of theoretical analysis, yes, the theoretical analysis presented in Section 4 is based on linearized error dynamics under the assumption of small perturbations. This assumption enables a tractable derivation and a clear understanding of how direct and indirect correction mechanisms differ in their amplification behavior. To address this theoretical limitation, we have extended our experimental setup to include exactly the kinds of challenging scenarios the reviewer refers to, like chaotic and turbulent cases. Results show that **INC maintains long-term stability with high accuracy across all test cases, which aligns with our theoretical analysis**. The alignment between our theoretical analysis and experimental outcomes reinforce our confidence in both the design and applicability of the INC framework.
>
> >W3: The hybrid approach relies on the differential solvers. However, for many industry applications, the legacy solvers might not be differentiable, which will limit the implementation of the proposed approach.
>
> We appreciate the reviewer’s comment, which correctly points out that the implementation of hybrid methods such as INC requires access to differentiable solvers. We acknowledge that many legacy solvers used in industry and scientific computing were not designed with differentiability in mind, which may indeed pose integration challenges for training paradigms that rely on gradient-based optimization. We will extend the corresponding discussion in future revisions. We also note that building differentiable numerical solvers is gaining momentum across multiple frameworks, such as JAX, PyTorch, and TensorFlow. We believe the development and adoption of differentiable solver backbones is a promising opportunity for deeper integration of machine learning into scientific computing workflows.
>
>
> ---
> References:
>
> [1] Kochkov, D. et al. Machine learning–accelerated computational fluid dynamics[J]. Proceedings of the National Academy of Sciences, 2021
>
> [2] Um, K. et al. Solver-in-the-loop: Learning from differentiable physics to interact with iterative pde-solvers[J]. Advances in neural information processing systems, 2020
>
> [3] List, B. et al. Differentiability in unrolled training of neural physics simulators on transient dynamics[J]. Computer Methods in Applied Mechanics and Engineering, 2025

---

> > ### Comment · Reviewer_8Ecs · 2025-08-05
> >
> > The rebuttal addresses my questions in detail and clarifies the motivation behind key design choices, such as multi‑step unrolling and stability considerations for stiff PDEs. However, the method still requires access to differentiable solvers, which limits applicability in industrial contexts where legacy solvers dominate. I keep my scoring.

---

> > > ### Author Response · Authors · 2025-08-06
> > >
> > > Thank you very much for your comments and acknowledgment. While relying on differentiable solvers does certainly introduce a difficulties for practitioners, we'd like to point out that our method at the same time makes important steps towards practical applications: e.g., we demonstrate it with spatially adaptive meshes, and show improvements for highly turbulent flows in 3D. Given the current trajectory of increased support for differentiability in solvers, we believe such examples are important to highlight the promise of the approach for industrial applications.

---

### Official Review · Reviewer_5gua · 2025-07-03

**Clarity:** 3
**Significance:** 3
**Originality:** 2
**Rating:** 5
**Confidence:** 3

**Summary:**

The work introduces Indirect Neural Correctors (INC), a new method of embedding neural networks within traditional numerical solvers (hybrid solvers). INC learns a correction of the form (with abuse of notation): $u_{t+1} = \mathcal{T}[u_{t} + G_{\theta}(u_{t})]$ where $\mathcal{T}$ is the temporal integration and $G_{\theta}$ is a neural network parameterized by $\theta$. The authors also include a theoretical analysis of INC to show that INC reduces errors proportional to $\Delta t^{-1} + L$ where $L$ is the Lipschitz constant and improves trajectory stability.

**Questions:**

1. See weaknesses 1. How is the work different from LC in [1, 2]?
2. See weakness 2. Can the authors include comparison to recent learned correction methods? I point to [1, 2] since I know they have code available.
3. What does SITL* mean? It is not clear to me what “a pre-correction variant” means. Similarly, can the authors provide more details on what the RNN baseline is?
4. How are $R^2$ scores being computed?

**Ethical Concerns:**

["NO or VERY MINOR ethics concerns only"]

**Final Justification:**

I already believed the paper was technically strong and of interest both to the Neural PDE solvers community and broader ML community after my initial review. My main concerns revolved around 3 topics: (1) clarification of contributions of the paper (2) unclear baselines and (3) details on R^2 and error metrics they use.

1) was sufficiently addressed in the rebuttal.

2) The authors clarified their method, each of the baselines, and added CSM (or reference [2] in my original review).

3) The response lays out clearly how the metrics are computed.

As a result of all my concerns being addressed, I'd like to raise my score to a 5.

**Limitations:**

yes

**Quality:**

3

**Strengths And Weaknesses:**

Strengths:
- The paper is well written and properly motivated. The problem being tackled is significant to the ML for dynamical systems research community.
- The proposed method shows strong improvements over the evaluated baselines and tests a variety of different dynamical systems.
- I believe the theoretical analysis is strong.

Weaknesses: There are two primary concerns I have with the paper.
1. To me, this seems very similar to [1, 2], whom the authors cite. The learned correction (LC) variant from both papers learns a correction of the form  $u^{n+1}=u^{n^*} + LC({u^{n^{\star}}})$. This seems to be similar to INC when rolled out: $u^{n+2} = \mathcal{T}[u^{(n+1)^\star}] + LC(u^{(n+1)^\star})= \mathcal{T}[u^{n^\star} + LC(u^{n^\star})] + LC(u^{(n+1)^\star})$
Here, the LC implicitly goes through the temporal integration step, albeit one timestep later. While I don’t think these are entirely the same, I would appreciate some further discussion on what makes INC uniquely different from LC [1, 2].
2. In a related vein, the baselines the paper evaluates against are weak compared to the hybrid solvers cited in the related works. Architecturally, I think the baselines (FNO, DeepONet, ResNet, U-Net) are sufficient. However, the STIL baselines do not, as far as I can tell, correspond to [1, 2] which would act as a more direct point of reference.

[1] Dmitrii Kochkov, Jamie A Smith, Ayya Alieva, Qing Wang, Michael P Brenner, and Stephan Hoyer. Machine learning–accelerated computational fluid dynamics. Proceedings of the National Academy of Sciences, 118(21):e2101784118, 2021
[2] Gideon Dresdner, Dmitrii Kochkov, Peter Norgaard, Leonardo Zepeda-Nunez, Jamie A Smith, Michael P Brenner, and Stephan Hoyer. Learning to correct spectral methods for simulating turbulent flows. arXiv preprint arXiv:2207.00556, 2022.

---

> ### Author Rebuttal · Authors · 2025-07-30
>
> We sincerely thank the reviewer for the constructive and thoughtful feedback. We appreciate the recognition of our theoretical analysis, broad empirical evaluation, and the significance of the problem within the ML for dynamical systems community. Your positive assessment of the clarity and relevance of our work is highly encouraging.
>
> We also carefully acknowledge the concerns raised, especially regarding the difference between our INC framework and prior learned correction (LC) approaches [1,2]. We would like to clarify that LC is actually included as a baseline in our paper, where we denoted it as SITL. We have now also implemented the method from [2], referred to as CSM, and compare to it in the following sections. Furthermore, we provide clarification on this comparison, as well as responses to your additional questions regarding our baseline variants (SITL*, RNN) and the computation of the $R^2$ metric.
>
> ## Questions
>
> >W1 & Q1: This seems very similar to [1, 2], whom the authors cite. The learned correction (LC) variant from both papers forms a correction of the form $u^{n+1} = u^{n} + \text{LC}(u^{n})$. This seems to be similar to INC when rolled out: $u^{n+2} = T[u^{n+1}] + \text{LC}(u^{n+1}) = T(u^{n} + \text{LC}(u^{n})) + \text{LC}(u^{n+1})$. The LC implicitly goes through the temporal integration step, albeit one timestep later. The reviewer wonders what makes INC uniquely different from LC [1, 2].
>
> We first want to address the reviewer’s concern about the similarity with the learned‐correction (LC) approach. In our paper, the baseline **SITL is actually the same as LC**; both correct the solution directly. In principle, the LC/SITL method works as follows. A coarse numerical solver $\mathcal{T}$ firstly advances the state, producing an intermediate solution $ u_{n}^{\*}=\mathcal{T}(u_n)$. Then a neural network $G_\theta$ **directly corrects** that solver output to yield the next state: $u_{n+1} = G_\theta\bigl(u_{n}^{\*}\bigr)$, where a residual connection has been applied: $G_\theta\bigl(u_{n}^{\*}\bigr)=u_{n}^{\*}+\theta(u_{n}^{\*})$. The equivalence of LC and SITL is an important point, which we will clarify in future revisions.
>
> We have found that when the correction is directly added to the solution, the simulation becomes more prone to perturbations. Any noise from the network will be directly added to the solutions. This has a stark effect on autoregressive forward simulations, where tiny errors will exponentially grow, especially for chaotic systems.
>
> On the contrary, when the correction is added to the right-hand side of the control equation, it affects the solution indirectly by going through the numerical integration. This process scales the error down by the Error Dominance Ratio as defined in definition 4.2 in the paper. In general, INC aims to nudge the solution back towards the correct trajectory at each time step indirectly. Your intuition is correct - if we consider a single step, the difference isn't obvious. Our results also show such a trend, where the difference between LC/SITL and INC is marginal in the initial stage. Figure 1 illustrates how LC/SITL and INC perform almost identically for the first 2k steps for the network based on FNO. However, INC shows stronger stability with accurate forward simulation for much longer horizons. Similar results can be observed in Figure 3 and Figure 6 as well.
>
> >W2 & Q2: In a related vein, the baselines the paper evaluates against are weak compared to the hybrid solvers cited in the related works. Architecturally, the FNO (Fourier Neural Operator), DeepONet, ResNet, U-Net are sufficient. However, the SITL baselines do not, as far as the reviewer can tell, correspond to [1, 2], which would act as a more direct point of reference.
>
> We thank the reviewer for the valuable suggestion about the baseline comparison with [1,2]. Our SITL implementation coincides with the LC [1] approach and differs only slightly from the variant presented in [2], **which we have implemented now, and denote as CSM**. To provide a more direct comparison, we have run additional experiments on both the Kuramoto–Sivashinsky (KS) and Backward‐Facing Step (BFS) cases. We evaluate each method with the same metrics, $R^2$ for KS and MSE for BFS, which align with the manuscript. Conceptually, CSM shares the same spirit as LC: it applies the learned correction directly to the coarse solver’s output, simply scaling the correction by the time‐step $\Delta t$. As a result, CSM remains susceptible to perturbation growth, particularly in chaotic regimes where small errors can rapidly magnify over long‐term rollouts. The quantitative results in Table 1 confirm **INC’s superiority across network architectures on the KS system and in the BFS flow**. For the KS problem, INC achieves $R^2$ scores markedly higher than the corresponding CSM results, and a roughly 42\% relative increase in $R^2$ has been observed with the DeepONet model.
>
> In the BFS case, the flow separation and reattachment zones are inherently unstable, where perturbations can trigger significant changes in such zones by affecting the main flow and the shear layer. INC attains an MSE of $(0.24\pm0.07)\times10^{-3}$, compared to $(8.09\pm1.52)\times10^{-3}$ for CSM, **an improvement in error magnitude by a factor of approximately 33**.
>
> These results demonstrate that INC outperforms LC (SITL in the manuscript) and also significantly improves upon variants like CSM [2]. This reinforces the effectiveness of our indirect correction strategy.
>
> Table 1: $R^2$ scores for KS (larger is better):
> ||KS(FNO)|KS(DeepONet)|KS(UNet)|KS(ResNet)|
> |-|-|-|-|-|
> |INC|0.97 $\pm$ 0.08 | 0.91 $\pm$ 0.14 | 0.97 $\pm$ 0.06 | 0.93 $\pm$ 0.13 |
> |CSM|0.90 $\pm$ 0.17 | 0.64 $\pm$ 0.34 | 0.89 $\pm$ 0.19 | 0.72 $\pm$ 0.32 |
>
> Table 2: MSE for BFS (lower is better):
> ||BFS(CNN)|
> |-|-|
> |INC| (0.24$\pm$0.07) $\times 10^{-3}$|
> |CSM| (8.09$\pm$1.52) $\times 10^{-3}$|
>
>
> >W3: What does SITL* mean? It is not clear to me what "a pre-correction variant" means. Similarly, can the authors provide more details on what the RNN baseline is?
>
> In response to the reviewer’s question, we provide additional details for SITL* and the RNN. SITL* is a simple pre‑correction variant of LC/SITL. Instead of adding the learned correction after the solver update, SITL* applies the neural term before the solver step, thus $u^{n+1}=\mathcal{T}(u^{\*}),\quad u^{\*}=G_\theta\bigl(u_n\bigr)=u^n+\theta(u^n)$.
>
> By evaluating both SITL and SITL*, we ensure our work covers the landscape of direct methods. Thus, the results demonstrate that regardless of ordering, any direct‑update method suffers from the same underlying error amplification $R_k$.
>
> Furthermore, the implementation of the RNN follows the architecture in the FNO paper[3]. It takes a sequence of past solution states $(u_{n-T+1}, \dots, u_n)$ and predicts the next state $u_{n+1}$ while operating autoregressively. Herein, following FNO, we set $T=10$. In contrast to the other approaches, the RNN is a solver-free and fully data-driven transition operator; other setups are kept identical to a hybrid solver, relying on learned temporal dependencies through autoregression. The comparison results reveal the limitations of these pure neural network architectures.
>
> >W4: How are $R^2$ scores being computed?
>
> The calculation of $R^2$ is defined as follows:
>
> $R^2(t)=1 -\frac{\sum_{d=1}^D \bigl(y_{t,d}^\text{true} - y_{t,d}^\text{pred}\bigr)^2}{\sum_{d=1}^D \bigl(y_{t,d}^\text{true} - \bar y_t^\text{true}\bigr)^2}
> ,\quad \text{where} \quad\bar y_t^\text{true} = \frac1D\sum_{d=1}^D y_{t,d}^\text{true}$.
>
> We thank the reviewer for pointing this out. The above definition will be explicitly included in the camera-ready version to improve clarity and reproducibility.
>
> ---
> References
>
> [1] Kochkov, D. et al. Machine learning–accelerated computational fluid dynamics[J]. Proceedings of the National Academy of Sciences, 2021
>
> [2] Dresdner G, et al. Learning to correct spectral methods for simulating turbulent flows. arXiv, 2022.
>
> [3] Li Z, et al. Fourier neural operator for parametric partial differential equations ICLR, 2021.

---

> > ### Comment · Reviewer_5gua · 2025-08-04
> >
> > Thank you for the clarifications and adding additional baselines. I will adjust my score accordingly.

---

### Note · Authors · 2025-08-13

Dear Reviewers and ACs,

We thank you for the thoughtful feedback of the reviewing phase, and want to highlight the core contributions of our work below:

*  **A Novel Mechanism:** We propose embedding the neural correction within the PDEs, which, as we prove theoretically, dampens error amplification over long-term rollouts compared to existing direct correction methods.

*  **Comprehensive Validation:** We show the effectiveness and generality of INC across six challenging 1D, 2D, and 3D tasks, using three different differentiable solvers and four distinct NNs, and diverse boundary conditions.

In our rebuttal, we have addressed the concerns of the reviewers, the most significant points being:

* We clarified that SITL = LC, SITL\* being its pre-correction variant. We added **CSM** as another hybrid solver baseline. **INC consistently outperforms CSM** (e.g., MSE of the BFS case were **33×** lower).

* We further **integrated the Modern-UNet**: INC reduces error by **\~70–84%** vs. LC/SITL, and clearly beats the pure-prediction baseline.

* We clarified fundamental differences to **P2C2Net**, which replaces the numerical scheme with a learned kernel and was not demonstrated for cases like our 3D turbulence scenario.

*   **Evaluation with Physical Metrics:** We have added new, physics-based metrics to our evaluation. INC more accurately captures the **energy spectrum, drag coefficient, and skin-friction coefficient** than direct correction methods (e.g., 72% lower MSE on skin-friction).

Thank you for taking the time to evaluate and consider our submission. We are looking forward to your final assessment.

Authors of submission 17912

---

### Decision · Program_Chairs · 2025-09-17

**Decision:**

Accept (poster)

**Comment:**

The authors proposed a method to couple a neural network correcting term with a traditional numerical solver. In case of a standard approach to correction a neural network directly corrects the solver output to yield the next state. As a result any noise from the network will accumulate and amplify during the autoregressive calculation of the equation solution. In case of the proposed method the correcing term is added to the right hand side of the control equation, so that the correction influences the solution indirectly through the numerical integration. The authors demonstrated both analytically and empirically that this indirect approach is more stable and provides more accurate results.

As a strengths of the paper I would highlight
- the paper is well written and properly motivated
- the experimental section convincingly demonstrates that the proposed approach outperforms other correcting schemes
- the authors derived an error-propagation framework for hybrid neural solvers and proved that the proposed indirect correction scheme reduces error compared to direct methods

As a weaknesses I would highlight
- the approache requires usage of differentiable solvers
- the theoretical analysis was conducted only for the backward Euler case and linearized error evolution

The main reasons for the accept decision are
- the authors proposed a simple but efficient framework to perform indirect neural correction of e PDE solution
- the experimental verification is detailed, the authors considered six challenging problems. Moreover, during rebuttal the authors also provided additional ablation experiments in response to the reviewers questions
- the method has some theoretical justification. Although the proof is provided under somewhat limited assumptions, the experimental verification confirmed that the method works under very broad conditions.

During the discussion with the reviewers several issues were raised that should be addressed in the final version of the paper. Here is the list of the main issues
- reviewer 5gua asked to clarify better differencies and similarities between direct and indirect schemes, and also asked to provide some new baselines
- reviewer Gzsb proposed to include several new baselines
- reviewer MzgN asked to provide more detailed explanation of the learning dynamics
- in principle, all reviewers asked relevant questions and the authors clarifications should be included in the main text or the appendix of the paper as they will help readers to position the approach better